# Predicting performance of naïve migratory animals, from many wrongs to self-correction

James D. McLaren [1✉], Heiko Schmaljohann[2,3] & Bernd Blasius [1,4]

Migratory orientation of many animals is inheritable, enabling inexperienced (naïve) individuals to migrate independently using a geomagnetic or celestial compass. It remains unresolved how naïve migrants reliably reach remote destinations, sometimes correcting for orientation error or displacement. To assess naïve migratory performance (successful arrival), we simulate and assess proposed compass courses for diverse airborne migratory populations, accounting for spherical-geometry effects, compass precision, cue transfers (e.g., sun to star compass), and geomagnetic variability. We formulate how time-compensated sun-compass headings partially self-correct, according to how inner-clocks are updated. For the longest-distance migrations simulated, time-compensated sun-compass courses are most robust to error, and most closely resemble known routes. For shorter-distance nocturnal migrations, geomagnetic or star-compass courses are most robust, due to not requiring nightly cue-transfers. Our predictive study provides a basis for assessment of compass-based naïve migration and mechanisms of self-correction, and supports twilight sun-compass orientation being key to many long-distance inaugural migrations.

[1] Institute for Chemistry and Biology of the Marine Environment (ICBM), University of Oldenburg, 26129 Oldenburg, Germany. [2] Institute for Biology and Environmental Sciences (IBU), Carl von Ossietzky University of Oldenburg, 26129 Oldenburg, Germany. [3] Institute of Avian Research, 26386 Wilhelmshaven, Germany. [4] Helmholtz Institute for Functional Marine Biodiversity (HIFMB), University of Oldenburg, 26129 Oldenburg, Germany. ✉email: james.mclaren@uol.de

Seasonal animal migrations have evolved across taxa at spatial scales spanning meters to continents[1]. A critical yet unresolved factor for migratory populations is how inexperienced (hereafter, naïve) individuals can perform inaugural migrations through unfamiliar habitats in unpredictable conditions. Experienced migrants are thought to perform true navigation, i.e., access a map sense to estimate the direction to migratory destinations[2–4]. Among migratory aquatic and terrestrial taxa, migration routes are often transmitted culturally by experienced cohorts, through collective and social cues[5,6]. However, many naïve airborne migrants complete their journeys to population-specific remote destinations (hereafter, goal areas) independently[2,7]. Among long-distance migrants, this is typically achieved in sequences of directed daily or nightly flights (hereafter, flight-steps), interspersed by periods of extended stopover (hereafter, stopover)[7–9]. Independently travelling naïve migrants are thought to accomplish such feats by following inherited migratory directions, re-determined at the onset of each flight-step using various geophysical migratory compasses[2,10]. However, the extent to which such compass courses (often termed clock and compass migration in the literature) can reliably reproduce observed migration patterns remains uncertain[11–14].

Based mainly on captive individuals, naïve but migration-ready birds and insects have been shown to orient consistently relative to both geomagnetic and celestial directional cues[2,10,15]. For example, migrating birds can innately identify the North–South geomagnetic axis, and distinguish magnetic North (N) from South (S) using geomagnetic inclination (the vertical tilt of the geomagnetic field, downward in the N Hemisphere). Unlike with the geomagnetic compass, the ability to maintain preferred directions relative to the sun and stars needs to be learned prior to migration[6,7]. The avian star compass identifies geographic N or S via the centre of celestial rotation (15° per hour clockwise), but does not respond to natural or experimental clock-shifts, i.e., is not time-compensated[16,17]. However, even outside the realm of migration, many insects[18,19] and birds[20,21] use a time-compensated sun compass, achieved by tracking the sun's azimuth angle, i.e., the horizontal projection of the sun's daily arc in the sky (earlier theories proposing bird navigation based on the sun's altitude are not supported[16,21]). Time-compensation based on sun azimuth would involve differential angular rates of adjustment throughout the day (e.g., is fastest at noon), and also vary across seasons and latitudes[22,23]. However, solar cues can potentially function as a time-limited compass, i.e., across shorter periods within the day[20]. A potential advantage of time-compensation close to sunrise or sunset is that the sun's azimuth then moves across the horizon at nearly the same angular speed throughout the year (i.e., only varies with latitude)[22]. With the sun and stars often being obscured by weather or topography, it is perhaps unsurprising that many airborne migrants can also orient using patterns of polarized light[24,25], which is less obscured by clouds[15,26]. The bands of maximal intensity of polarized light are in fact perpendicular to sun azimuth during both sunrise and sunset, which if averaged has been proposed as a way for migrants to identify the geographic N–S axis[24,27].

Given the diversity and complexity among compass cues, together with confounding factors such as wind and topography[7,9,28,29], it is not surprising that little is known regarding which cues are used in flight, or whether cue use varies across entire routes[2,30,31]. Prior to departure, night-migratory songbirds seem to prioritize one (hereafter, primary) compass system to determine flight-step headings, sometimes transferred to a second, in-flight compass[10,30,31]. Cue-conflict experiments suggest various contingencies and hierarchies involving calibration between compasses, but often prioritization of celestial cues at twilight, particularly among North American migrants[10,15,31].

The choice of primary compass can result in substantially different compass courses, with five main classes proposed: geographic loxodromes, geomagnetic loxodromes, magnetoclinic courses, fixed sun compass and time-compensated sun compass courses. Geographic loxodromes follow constant headings relative to the geographic N–S axis, which is potentially identifiable using either a primary star compass[16,30] or by averaging polarized light cues at sunrise and sunset[24,27]. Geomagnetic loxodromes follow constant headings relative to the proximate geomagnetic N–S axis, resulting in an offset to geographic headings according to proximate geomagnetic declination[10,32]. Magnetoclinic courses shift gradually and increasingly towards the South (N in the S Hemisphere), by maintaining a fixed (transverse) projection of proximate geomagnetic inclination en route[33,34]. With sun compass courses, flight-step headings are determined relative to proximate sun azimuth, here focused on sunset courses (sunrise courses for day-migrating species)[15,34]. Fixed sun compass courses follow a constant heading relative to sunset azimuth, resulting in less consistent orientation shifts with date and location, e.g., asymmetrically between Eastward and Westward courses[15,34]. A migrant following a time-compensated sun compass (TCSC) course also orients relative to proximate sunset azimuth, but after crossing longitudes, its heading on the subsequent flight-step becomes clock-shifted[22]. This plausibly occurs between consecutive flight-steps (i.e., without extended stopovers), given the avian inner-clock apparently requires several days to adjust[21]. TCSC sunrise or sunset courses shift more consistently Southwards (Northwards in the S Hemisphere) compared with fixed sun compass courses, resulting in close to great circle trajectories[22,35].

The extent to which compass courses can result in successful migration routes remains an open question. Central to this question is whether cue perception, compass headings and resultant flight-step directions are sufficiently precise, accounting for overall directional errors being reduced over many flight-steps, known as a many-wrongs effect[5,36,37] (we may assume that population-mean migratory headings are accurate, i.e., maximize the probability of arrival at goal areas). Among the avian compasses, the magnetic compass has been estimated to be as precise as 0.5°[38,39], and sun compass within 5°[21], with variability of night-migratory headings aloft to be 20°–30°[40,41]. However, recoveries of ringed juvenile songbirds suggest upwards of ~50° directional variability among flight-steps (circular length 0.665)[36]. Simulated fixed-heading (loxodrome) migration on a plane indicates that such variability is only compatible with migration along very broad fronts[12,14,36]. Some directional variability among recovered migrants may at least in part be negotiable independently of a compass-course process[42] (e.g., post-fledging dispersal[43], regional-scale stopover movements[44], and responses to wind and topography[9,28,41]). Nonetheless, migratory tracking data reveal a more diverse picture than constant-heading loxodrome movements, often featuring narrow movement corridors, and both gradually and sharply direction-changing routes[11,14]. While simulations of known bird-migration routes on the sphere often resemble direction-shifting TCSC or magnetoclinic courses[33,35], their relative feasibility has been debated[34,45,46], and their robustness to errors remains untested. Moreover, the modulation of orientation errors by spherical-geometry effects remains unquantified for compass-based animal movements[36,47].

One way to improve the robustness of compass-based movement would be if animals possessed a self-correction mechanism. Remarkably, some naïve bird migrants have been shown to adjust their orientation correctively following either natural or experimental displacement[42,48–50]. The ability to correct orientation following longitudinal displacements is a hallmark of true navigation[2,3], but could also be achievable if migrants tracked

configurations of stars[16,17] or the sun[16] in a time-compensated way (also termed *pseudo-navigation*[42]). While such corrections are thought to be small for migration-relevant displacements[16,42], their cumulative effect over many flight-steps and entire routes have not been assessed, nor have they been quantified regarding time-compensated responses to sun azimuth.

Here we provide a modelling framework to assess factors governing the performance and robustness of airborne migration for compass-based movement on a sphere. For simplicity and interpretability, we focus on compass courses based on a single inherited or imprinted heading. We quantified migratory performance as fractional successful arrival within goal areas and assessed robustness among compass courses both algebraically as sensitivity to the error between successive flight-steps, and by performance across entire routes. For the latter, we developed a spatiotemporal migration model to simulate each compass course over a broad range of errors, for both a generic migrant globally and for diverse airborne migratory species and routes incorporating dynamic geomagnetic data[51]. To facilitate the algebraic and global generic-migrant simulations, we additionally modelled geomagnetic courses for a dipole Earth, where inclination varies solely with magnetic latitude, which explains 90% of the Earth's magnetic variation[32]. To ensure consistent sun compass trajectories, we assumed that flight-step headings on commencement of migration were imprinted from inherited geographic or geomagnetic headings[2,10,30]. We also extended existing formulations of TCSC to assess critical assumptions regarding how inner clocks are reset and courses are maintained in variable flight and stop-over schedules. Regarding orientation errors, in addition to considering directional precision among daily or nightly flight-steps, with sources of error implicit, we also considered biologically relevant sources of error within flight-steps, quantified by compass precision (governing initial cue detection, possible cue transfers to a secondary compass, and in-flight cue maintenance), within-flight drift (through compass bias or wind), and between-individual variability in inherited headings[12]. Finally, based on our formulations and species simulations, we identified flight capacity and route-geometric factors affecting compass course performance on the sphere and assessed their relative effect among compass courses and migration routes using nonlinear regression and AICc model selection[52].

## Results

Table 1 lists terms relating to stepwise movement, compass cues, orientation precision and proposed compass courses (see also the "Methods" section and Supplementary Information). Before describing sensitivity and performance among compass courses, we briefly outline stepwise movement on a sphere, modelling of circular headings and error, and expected migratory performance assuming a planar Earth, including the effect of precision among and within flight-steps.

**Stepwise movement on a sphere.** For a sequence of $N$ daily or nightly flight-steps ($i = 0, \ldots, N-1$) with corresponding flight-step directions, $\alpha_i$, clockwise relative to geographic S (anticlockwise to N in the S Hemisphere), the latitude, $\varnothing_{i+1}$, and longitude, $\lambda_{i+1}$, following each flight-step can be approximated as

$$\phi_{i+1} = \phi_i - R_{\text{step}} \cos\alpha_i, \tag{1}$$

$$\lambda_{i+1} = \lambda_i - R_{\text{step}} \sin\alpha_i / \cos\phi_i \tag{2}$$

where $R_{\text{step}}$ is the flight-step distance (with all arguments in radians). A key spherical-geometry factor is an initial latitude, $\varnothing_0$, which iteratively affects progress in longitude (Eq. (2)) and,

through the cosine factor in the denominator, magnifies the effect of any orientation errors disproportionately at high latitudes.

**Modelling circular headings and error.** Throughout the study, we determined headings following a von Mises distribution, the circular equivalent to the normal distribution, with angular precision quantified by the von Mises concentration, $\kappa$[53]. To facilitate interpretation, we also describe compass and flight-step precision in degrees, as well as variability in inherited headings, according to $\sigma = 1/\sqrt{\kappa}$, which closely resembles standard angular deviation for $\sigma \leq 30°$ ($\kappa > 3.7$, circular length $> 0.85$; see Supplementary Fig. 1 and Note 1)[53]. For error scenarios with multiple components, we also describe directional precision among flight-steps using the normal approximation $\sigma_{A+B} = \sqrt{\sigma_A^2 + \sigma_B^2}$ (which also becomes poorer for $\sigma > 30°$; see Supplementary Fig. 1 and Note 1).

**Migratory performance assuming a planar Earth.** Figure 1a illustrates stepwise compass-based movement across a distance $R_{\text{mig}}$ to a migratory goal area of radius $R_{\text{goal}}$. The probability of successful arrival will increase with increasing directional precision among flight-steps, and with two population and route specific factors: (1) goal-area breadth, quantified as the ratio of goal radius to migration distance, $\beta = R_{\text{goal}}/R_{\text{mig}}$, and (2) and following the many wrongs principle, with increasing number of required flight-steps, which in the error-free case we term $N_0$. In the normal limit (see the "Methods" section), for a given directional precision among flight-steps, performance among routes will vary according to what we term the length-adjusted goal breadth

$$\beta_{\text{adj}} = \sqrt{N_0} R_{\text{goal}} / R_{\text{mig}}. \tag{3}$$

**Precision among and within flight-steps.** If we consider a single flight-step based on a single compass cue, higher frequency of in-flight cue maintenance will reduce expected flight-step errors in a many-wrongs fashion (Fig. 1b). This increased directional precision comes at the expense of flight-step distance, but not extremely so for cue-maintenance precision within ~60° (Supplementary Fig. 1b, c). Contrastingly, for flight-steps involving cue transfer to a second compass, more frequent cue maintenance does not make up for initial cue detection and transfer errors (Fig. 1c, and see Supplementary Note 1). Therefore, within a single nocturnal flight-step, assuming equivalent precision and availability among cues, non-transferred geomagnetic or star-compass headings are relatively more precise compared with nightly flights transferred from a sun compass.

**Compass course formulations and sensitivity.** In the "Methods" section, we formulate flight-step headings for each compass course (see also Table 1), including magnetoclinic courses in a geomagnetic dipole. To assess sun-compass sensitivity algebraically, and also to improve computational efficiency, we used an algebraic expression for sunset azimuth as a function of latitude and day of the year (Eq. (9)). The heuristics of TCSC courses and self-correction are illustrated in Fig. 2. Following error-free headings, a migrant's subsequent heading will shift oppositely to its clock-shift, creating an increasingly Southward trajectory (Northward in the S Hemisphere)[34]. Following an imprecise heading and ensuing longitude error, $\Delta\lambda$, the difference in clock shift compared with the error-free case will tend to counteract the previous error. The expected self-correcting offset in heading, $\Delta\bar{\alpha}$, follows the same relationship as with TCSC courses in the error-

**Table 1 Definitions of terms describing stepwise movement, among and within-step precision, and geophysical orientation cues.**

| | Variable or factor | Description |
|---|---|---|
| Stepwise movement | Flight-step | Encompasses departure and (daily or nightly) flight (Fig. 1a). Identified by subscript $i$, also subdivided hourly to include within-step processes. |
| | Location | Flight-step latitude, $\varnothing_i$, and longitude, $\lambda_i$, in radians (Eqs. (1) and (2)). Geomagnetic-dipole simulations use geomagnetic latitude and longitude. |
| | Step length | Flight-step distance, $R_{step}$ (radians), constant or subdivided hourly. |
| Orientation and precision | Preferred heading | Expected heading, $\bar{\alpha}_i$, based on the primary compass. Defined clockwise from geographic South (S), counter-clockwise from N in S Hemisphere. |
| | Flight orientation | Quantified using von Mises distribution with concentration parameter, $\kappa$[53]. We describe orientation precision, and also between-individual variability in inherited headings, by $\sigma = 1/\sqrt{\kappa}$ ($\cong$angular std. deviation for $\sigma \leq 30°$, $\kappa > 3.7$). |
| | Compass precision | Within flight-steps, affects initial cue detection, if applicable cue transfer and cue maintenance, i.e., in-flight redetermination of headings (Fig. 1). |
| | Error scenarios | We modelled precision both among flight-steps (0°–60° precision, implicitly including all sources of errors) and considering biologically relevant within-step variability (0°–40° compass precision in cue detection, transfers and maintenance, up to 20° within-flight drift, and 2.5° default between-individual variability). |
| Geophysical orientation cues | Geomagnetic axis | Offset from geographic headings by magnetic declination, $\delta_m$[32] (constant in dipole model, otherwise interpolated from IGRF data[51]). |
| | Geomagnetic inclination | Angle of field vector to horizontal, $\gamma_i$. Strongly latitude-dependent[33,51]. |
| | Sun azimuth | Sunrise or sunset azimuth, $\theta_s$ (Eq. (9)). Along time-compensated sun compass courses, clock-shifted until resetting of inner-clock. |
| | Polarized light | At sunrise and sunset, maximum bands of polarized light are perpendicular to sun azimuth, and average to geographic N–S. |
| | Stellar axis | Fixed star or centre of rotation. Not time-compensated between steps[16,17]. |
| Compass courses | Geographic loxodrome | Constant heading relative to geographic axis, identifiable by a star compass, or by averaging polarized light cues between dawn and dusk[24]. |
| | Geomagnetic loxodrome | Constant heading relative to perceived geomagnetic axis. |
| | Magnetoclinic | Geomagnetic headings based on maintaining a fixed transverse projection of proximate inclination[33]. |
| | Fixed sun compass | Constant heading vs. sunrise or sunset azimuth. |
| | Time-compensated sun compass (TCSC) | As in fixed sun compass, but offset due to longitudinal clock-shifts, according to how migrants track sun azimuth[22]. We further extended the original formulation[22] to allow for proximate sun-azimuth tracking, and headings to be retained from the first night of extended stopover (rather than on arrival, which is less consistent with sun azimuth headings). |

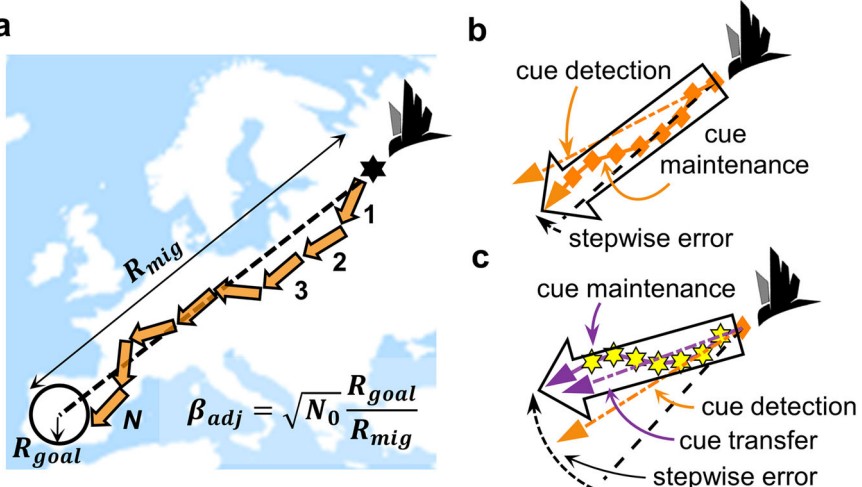

**Fig. 1 Compass-based movement, precision between and within flight-steps. a** Schematic of $N$ migratory flight-steps (orange arrows), based on a single preferred heading (dashed black line), spanning a distance $R_{mig}$ to a migratory destination ("goal area", with radius $R_{goal}$). For a given and sufficiently high precision among flight-steps, and ignoring spherical-geometry effects, the probability of successful arrival increases with goal area, the number of required error-free flight-steps, $N_0$, but decreases with migration distance (inset and Eq. (3)). **b** Within-flight compass precision based on a single (e.g., geomagnetic) cue. The expected initial error in cue detection (angle between dashed orange and black lines) will on average be offset by repeated, e.g., hourly cue maintenance within flight-steps (solid orange line and diamond shapes). **c** Contrastingly, with transfer to a secondary compass (dashed-purple line, e.g., star compass), the expected flight-step error will exceed cue-detection errors, regardless of cue maintenance (solid purple line and yellow hexagons). Bird icon from http://www.dreamstime.com (ID 16983354).

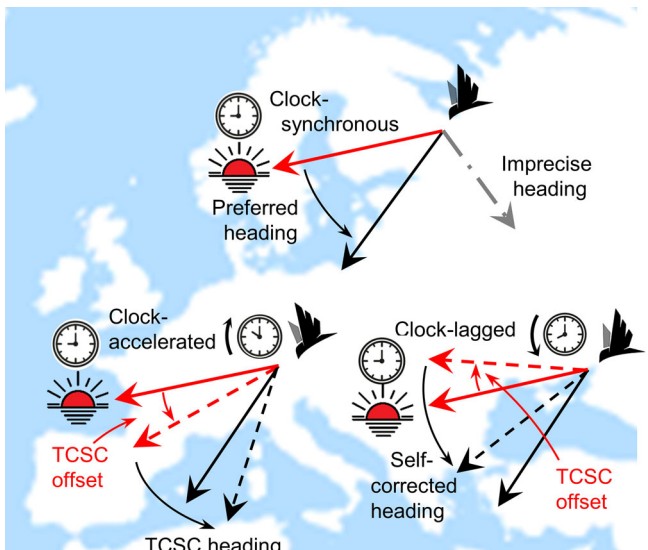

**Fig. 2 Time-compensated sun compass (TCSC) headings and self-correction.** A TCSC migrant clock-synchronized to local conditions (top) maintains its preferred direction (solid black arrow) by adjusting its heading relative to the daily clockwise rotation in sun azimuth (here at sunset, solid red arrow). Following an error-free flight-step (lower left), the longitudinally displaced migrant will be clock-shifted relative to local time. Here, the clock-accelerated shift results in an over-compensation to proximate sun azimuth, i.e., counter-clockwise TCSC offset (dashed red arrow), hence increasingly Southward heading (dashed black arrow). If the migrant's initial heading is imprecise (dot-dashed grey line), its longitudinal displacement will lead to a contrasting clock-shift. Here, the clock-lagged migrant (lower right) will under-compensate relative to proximate sun azimuth, resulting in a clockwise offset (dashed red arrow) and hence a self-corrected heading (dashed black line). Between-step shifts in proximate sunset azimuth become biologically relevant at multi-day and multi-step scales (Fig. 7). Images from www.dreamstime.com and www.flaticon.com.

free case (see the "Methods" section):

$$\triangle\bar{\alpha} \cong \triangle\lambda\sin\phi \qquad (4)$$

To assess the robustness of TCSC courses to variable migratory schedules, we additionally formulate in the "Methods" section (and illustrated below) the effects of changes in latitude, clock-resets, extended stopovers and angular speed of sun azimuth.

The sensitivity of successive headings to orientation errors contrasted strongly among compass courses, particularly at high latitudes and with significantly Eastward or Westward flight directions. The heat maps in Fig. 3 depict expected percentile growth (red) or self-correction (blue) of directional errors between successive flight-steps, with the arrows depicting how (error-free) headings for each compass course shift with latitude along prototypical routes. Per definition, preferred geographic loxodrome headings do not depend on previous headings, resulting in no expected growth or correction in error (Fig. 3a, Eq. (5)). This also holds for geomagnetic loxodrome headings in a dipole field, relative to geomagnetic axes (Eq. (6)). Contrastingly, the latitude-dependence of magnetoclinic headings (Eqs. (7) and (8)) renders them inter-dependent, and leads to extremely high sensitivity for virtually any non-Southerly heading (Fig. 3b, Eq. (13)). Fixed sun compass headings remain largely insensitive to errors, but at high latitudes will iteratively grow or self-correct (up to ~10°% in the sub-Arctic), depending on whether East or West oriented, and before or after the fall equinox (Fig. 3c, d, Eq. (14)).

Sensitivity in TCSC headings is similarly East–West antisymmetric about the equinox (Fig. 3e, f), but their self-correcting nature (Fig. 2) further reduces expected subsequent errors, with 5–25% self-correction at mid to high latitudes and over a broad range of directions (Eq. (15)), into which headings (blue arrows) moreover tend to converge. While the degree of TCSC self-correction remains small away from polar latitudes (as shown in Fig. 2, roughly to scale), subsequent steps will continue to self-correct for any remaining discrepancies in longitude until inner clocks are reset.

**Simulation of migration routes.** Using our migration model, we derived initial headings maximizing performance (successful arrival) for each modelled migratory population and error scenario. We considered both directional precision among flight-steps up to 60° ($\kappa = 0.9$, circular length 0.4) and biologically relevant scenarios resulting in a similar range in directional precision: 0°–60° compass precision (governing initial cue detection, cue transfers and cue maintenance), 0°–20° within-flight drift and 0°–10° between-individual variability in inherited headings. For illustrative purposes, trajectories are depicted for default error scenarios with 20° directional precision among flight-steps ($\kappa = 8.2$, circular length = 0.94), and for a biologically relevant error scenario with 15° compass precision and drift ($\kappa = 14.6$, circular length = 0.97) and 2.5° between-individual variability ($\kappa = 525$, circular length = 0.999). The default biologically relevant scenario results in directional precision among flight-steps of ~28° ($\kappa = 4.2$, circular length = 0.87) for cue-transferred and ~16° ($\kappa = 12.8$, circular length = 0.96) for non-transferred courses.

We simulated compass courses for seven night-migratory bird species, the Nathusius bat (*Pipistrellus nathusii*) and the daytime-migrant monarch butterfly (*Danaus plexippus*). Species are listed in Table 2 in increasing order of expected performance (length-adjusted goal breadth, Eq. (3)), together with other key model parameters including migration distance, goal area, migration period, great circle headings, stopover durations and travel speeds. For the night-migratory species, TCSC courses are cue-transferred and geographic loxodrome courses represent non-transferred star compass courses. For the daytime-migrant monarch butterfly, sun compass courses are non-transferred but geographic loxodromes assumed to be cue-transferred (from a star compass or daily-averaged polarized-light).

Compass-course performance among species varied overall as expected relative to length-adjusted goal-breadth, as illustrated in Fig. 4a for 20° precision among flight-steps. TCSC courses always performed best, with geomagnetic loxodrome courses being less consistent, and magnetoclinic courses performing overall worst. However, when accounting for precision within flight-steps, cue-transfer errors diminished the relative advantage of nocturnal TCSC courses. This is illustrated in Fig. 4b–f for five species with the default biologically relevant error scenario (including 15° compass precision), with the remaining species in Supplementary Fig. 2. With this scenario, TCSC courses outperform loxodromes for the routes requiring the largest number of flight-steps, i.e., the willow warbler (*Phylloscopus trochilus*), grey-cheeked thrush (*Catharus minimus*) and monarch butterfly, and also most closely matches their known routes (grey arrows). For the ca. 14,000 km willow warbler route and nearly West–East common rosefinch (*Carpodacus erythrinus*) routes, the default-error magnetoclinic courses were virtually infeasible despite their relatively high directional precision (~16°) among flight-steps. Cue-transferred courses are presented here based on a nocturnal star compass, but transfers to a geomagnetic in-flight compass perform overall very similarly (Supplementary Fig. 3).

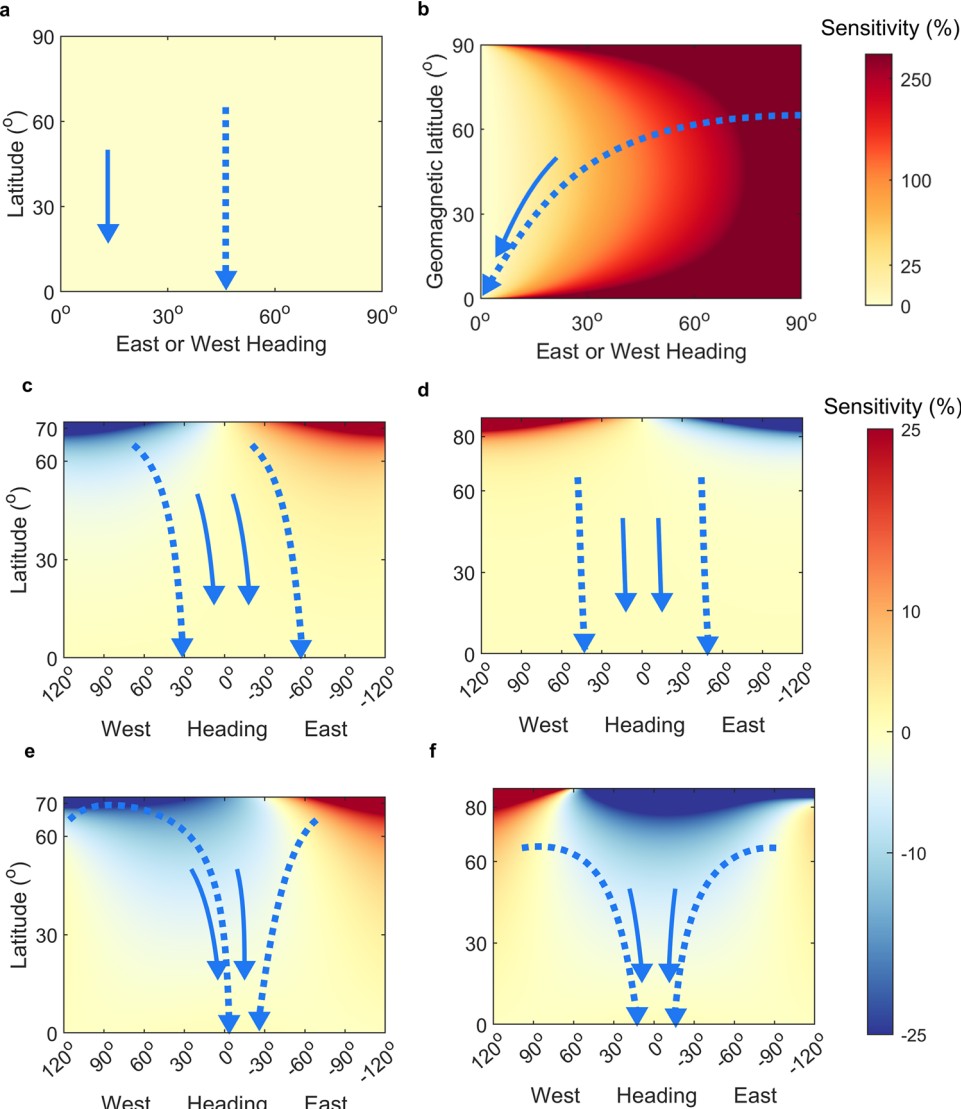

**Fig. 3 Contrasting sensitivity in headings among compass courses.** Sensitivity between flight-step headings, quantified as percentage growth of small errors between successive headings (colour scales on right), as a function of current heading (clockwise from South) and latitude (geomagnetic latitude for geomagnetic courses), for **a** constant-heading geographic loxodromes, or equivalently geomagnetic loxodromes in a geomagnetic dipole Earth, **b** magnetoclinic courses in a geomagnetic dipole, **c** fixed sun compass courses on August 1 and **d** October 1, and **e** time-compensated sun compass (TCSC) courses on August 1 and **f** October 1. For positive (yellow to red coloured) values of sensitivity, expected errors in the successive headings will grow iteratively, whereas, for negative (white to blue coloured) values, expected headings are self-correcting. Blue arrows depict error-free headings for travel from (solid lines) 50°N–15°N across 10° in longitude, and (dashed lines) 65°N–0°N across 90° in longitude. For all simulations, flight-step distances were 360 km. In **c–f**, regions without sunset or sunrise (poleward of ~72° on August 1 and ~87° on October 1, respectively) are not depicted.

Patterns and hierarchies in performance among compass courses were similar for the global simulations of a generic migrant to a goal with 500-km radius in a geomagnetic dipole (Fig. 5). For biologically relevant error scenarios with 30° or better compass precision ($\kappa > 3.7$, circular length > 0.85), TCSC courses were feasible across longer longitudinal migration distances compared with other courses (especially magnetoclinic courses; Fig. 5a), and also outperformed fixed sun compass and magnetoclinic routes across their limited ranges (Fig. 5b–e). However, particularly for migration at mid-latitudes, cue-transferred TCSC courses lose their self-correcting advantage relative to non-transferred loxodromes, which they only outperform across longer-distance routes and with compass precision within ~15° (Fig. 5d).

**Uncertainty analysis of migration parameters and formulation.** Compass course performance can vary widely depending on estimated model parameters[12,14]. Figure 6 illustrates the diverse effects of estimated between-individual variability (up to 10°, i.e., $\kappa = 33$, circular length = 0.985), within-flight drift and goal radius (100–1100 km) on migratory performance for the nearly N–S migration (Fig. 6a) of Finnish-breeding marsh warblers (*Acrocephalus palustris*) to East Africa[12,14]. Simulations were performed as per Table 2 and Supplementary Fig. 2, but with fixed migratory schedules. In the absence of drift effects (Fig. 6b), non-transferred geomagnetic loxodromes (orange line) and geographic loxodromes (purple line) outperform cue-transferred sun compass courses (dashed green line), unless the star compass is unavailable on departure (e.g., due to clouds), necessitating a cue transfer

**Table 2 Model parameters of the species compass course simulations.**

| Species (reference) | Route | Goal radius (km) | Date of departure (and last arrival) | Distance (km) | Initial heading (°) | Flight speed (m/s) | Flight-step distance (km) | Consecutive flight-steps, stopover duration (d) | Travel speed (km/d) | Minimum (maximum) flight-steps | Length-adjusted goal breadth |
|---|---|---|---|---|---|---|---|---|---|---|---|
| Monarch butterfly, *Danaus plexippus*[25] | Quebec–Mexico | 100 | Aug 15 ± 14 (~Dec 13) | 3290 (3300) | 219 (213) | 3.0 | 85 | 5, 3 ± 1 | 55 | 36.9 (77) | 0.18 |
| Ring ouzel, *Turdus torquatus*[78] | Scotland–N Africa | 250 | Aug 31 ± 7 (~Nov 29) | 2610 (2610) | 181 (179) | 11.5 | 330 | 10, 15 ± 5 | 130 | 7.1 (42) | 0.26 |
| Common rosefinch, *Carpodacus erythrinus*[79] | Bulgaria–NW India | 400 | Aug 7 ± 7 (~Nov 29) | 5110 (5170) | 96 (123) | 12.5 | 360 | 5, 5 ± 2 | 400 | 13.1 (44) | 0.28 |
| Marsh warbler, *Acrocephalus palustris*[12,14] | Finland–Kenya | 500 | Sep 1 ± 7 (~Jan 1) | 6720 (6730) | 168 (173) | 11.5 | 330 | 5, 5 ± 2 | 165 | 18.8 (48) | 0.32 |
| Kirtland's warbler, *Setophaga kirtlandii*[80] | Michigan–Bahamas | 300 | Oct 6 ± 7 (~Dec 5) | 2370 (2370) | 157 (160) | 10 | 290 | 5, 5 ± 2 | 145 | 7.2 (33) | 0.34 |
| Nathusius bat, *Pipistrellus nathusii*[81] | Latvia– Spain | 300 | Aug 15 ± 14 (~Nov 13) | 2040 (2050) | 233 (224) | 7.5 | 160 | 3, 5 ± 2 | 60 | 10.8 (36) | 0.48 |
| Willow warbler, *Phylloscopus trochilus yakutensis*[64] | Siberia–Zambia | 1000 | Sep 1 ± 7 (~Jan 1) | 13,200 (14,600) | 311 (233) | 10.5 | 300 | 5, 2 ± 2 | 215 | 40.2 (87) | 0.48 |
| Grey-cheeked thrush, *Catharus minimus*[54,55] | Yukon–Columbia | 1000 | Sep 10 ± 7 (~Jan 7) | 9080 (9300) | 108 (141) | 11.5 | 330 | 5, 5 ± 2 | 165 | 24.4 (63) | 0.54 |
| Eurasian hoopoe, *Upupa epops*[82] | Switzerland –W Africa | 800 | Aug 10 ± 7 (~Oct 9) | 3370 (3380) | 204 (200) | 12.0 | 345 | 5, 5 ± 2 | 170 | 7.4 (33) | 0.65 |

Routes and other parameters for each modelled species were based on tracking and other studies, including goal areas, initial departure dates ± standard deviation (and maximum arrival date), great-circle (followed by loxodrome) distances and headings, flight (ground) speed, travel (migration) speeds, and migration schedule, the latter modelled as a (fixed) sequence of consecutive flight-steps followed by an extended stopover (mean ± standard deviation). Species are listed in increasing order of expected migration performance in the normal planar limit, based on length-adjusted goal breadth, $\beta_{Adj}$ (Eq. (3)). All migrants except the monarch butterfly are principally night-migratory.

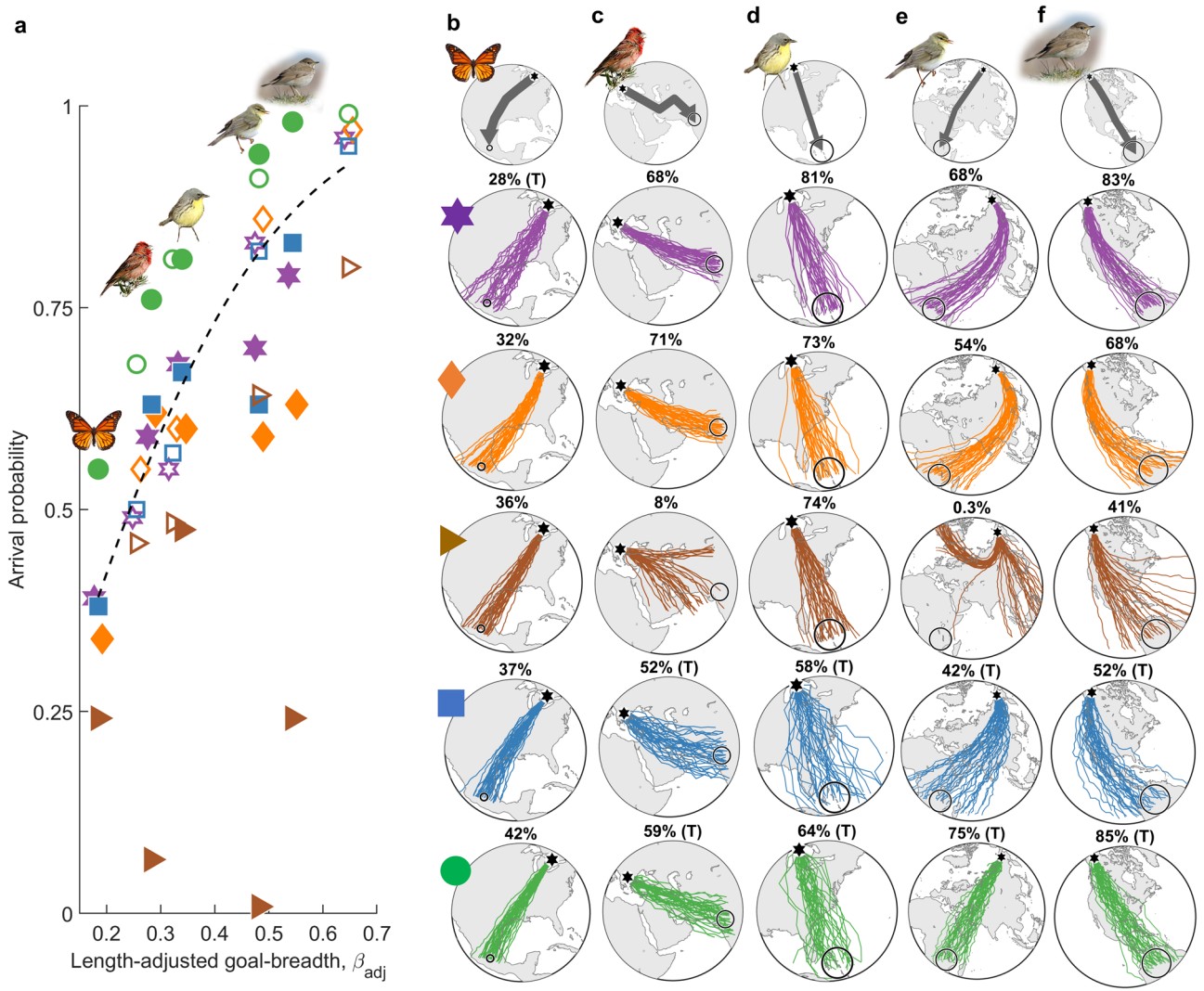

**Fig. 4 Diverse compass course performance among species and migration routes. a** Compass-course performance along known routes of nine airborne migrant species (Table 2) vs. length-adjusted goal breadth (Eq. (3)). Illustrated here based on 20° precision among flight-steps, with filled symbols representing (left-right) monarch butterfly, common rosefinch, Kirtland's warbler (*Setophaga kirtlandii*), willow warbler and grey-cheeked thrush, and open symbols representing the other species (depicted in Supplementary Fig. 2). Purple hexagons represent geographic loxodromes, orange diamonds geomagnetic loxodromes, brown triangles magnetoclinic courses, blue squares fixed sun compass courses and green circles time-compensated sun compass (TCSC) courses. **b–f** Randomly sampled trajectories (from 10,000 modelled individuals) with route-optimal population-mean headings for the above species, with colours and symbols representing compass-course as in **a**, for the above-named species (with the others depicted in Supplementary Fig. 2), assuming biologically relevant variability including 15° compass precision, drift, and 2.5° between-individual variability in inherited headings (see text). The top row depicts known species routes (grey arrows) between natal grounds (black hexagons) and natural goal areas (open circles), with straight lines appearing as great circles in the stereographic projection. Performance (percentage arrival) and, where applicable, also cue-transferred courses ("T") are depicted above each panel. Photos by **b** D. Descousens (https://creativecommons.org/licenses/by-sa/2.0/), **c** I. Shah (https://creativecommons.org/licenses/by-sa/4.0/), **d** B. Majoros (https://creativecommons.org/licenses/by-sa/3.0/), **e** HS and **f** A. D'Entrement.

(dashed purple line). Cue-transferred TCSC courses were however relatively less affected by 20° within-flight drift (Fig. 6c), even outperforming non-transferred loxodromes for compass precision within ca. 15° ($\kappa = 14.6$, circular length $= 0.97$). Figure 6d depicts the effects of goal radius (km) and within-individual variability on the performance of geographical loxodromes with 20° compass precision in the absence of drift. Figure 6e, f illustrate that the performance gain with TCSC courses over geographic loxodromes is larger with larger between-individual variability, and for larger goal areas, particularly in presence of (20°) within-flight drift (Fig. 6f).

The feasibility of TCSC courses across broad latitudinal distances depends on two critical assumptions[22] (see the "Methods" section): that (1) the temporal rate of time-

compensated orientation adjustments are updated and retained during extended stopover periods, and (2) geographic flight-step headings are retained on arrival at stopovers. In Fig. 7, we assess these assumptions for cross-continental grey-cheeked thrush migration[54,55], and explore possible alternative behaviours. Simulations otherwise followed Table 2 but with double the variation in initial migration date (indicated by trajectory colour). Classic TCSC trajectories, without extended stopovers or resetting of inner clocks (Fig. 7a), resemble both great circles and known routes (grey arrow in inset). This, however, relies on nightly departure headings being adjusted according to the angular speed of sun-azimuth experienced on the natal grounds. Contrastingly, when adjusting nightly headings to proximate angular speeds of sun-azimuth (Fig. 7b), trajectories deviate strongly from great

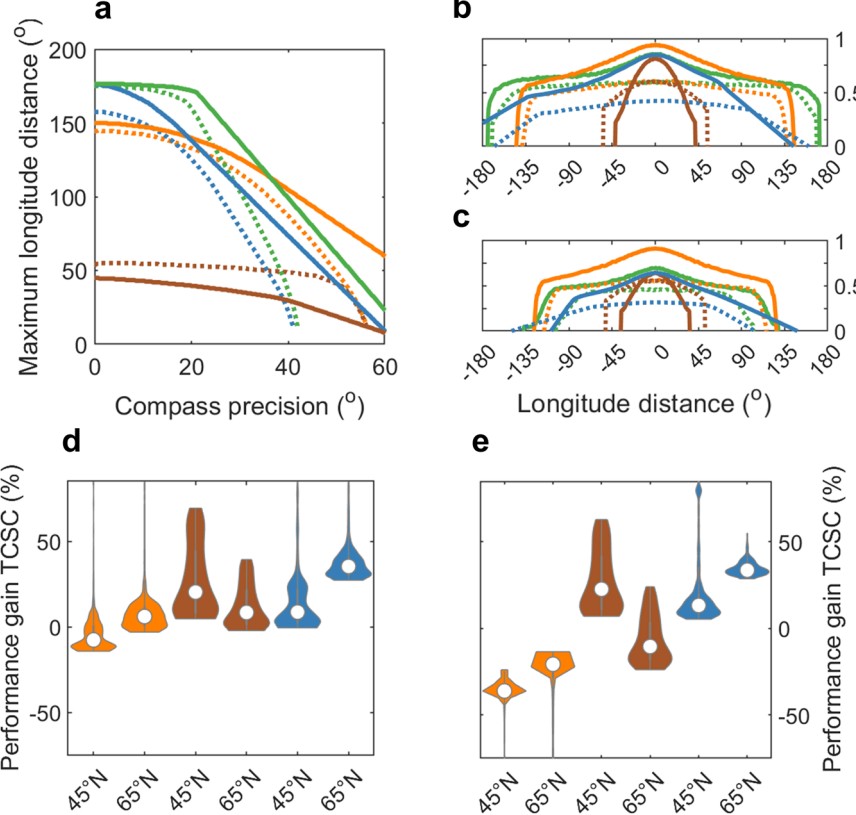

**Fig. 5 Global migration ranges and compass-course performance for a generic migrant.** Feasible longitude migration distances and relative performance among compass courses based on simulations for a generic nocturnally migrating species in a geomagnetic dipole Earth, with biologically relevant error (0°–60° magnitudes in compass precision, and 15° drift). **a** Maximal longitudinal migration distances (with at least 25% success of arriving within 500 km of a goal) vs. compass precision (both in degrees), with line colours among compass courses as in Fig. 4 (orange: geomagnetic loxodrome, brown: magnetoclinic, blue: cue-transferred fixed and green: cue-transferred TCSC courses). Solid lines represent migration between 45°N–25°N and dotted lines between 65°N–0°N. **b** and **c** Performance among compass courses with compass precisions of **b** 15° and **c** 30°, with solid lines representing migration between 45°N and 25°N and dotted lines between 65°N–0°N. **d** and **e** Violin plots, with violin widths depicting distributions of percentage gain with TCSC relative to other courses among all feasible longitudinal distances for routes between 45°N and 25°N and between 65°N and 0°N, with colours matching **a**, and for compass precisions of (**d**) 15° and (**e**) 30°.

circles. This can be averted when (Fig. 7c, *sensu* Alerstam[22] and as in Fig. 4) geographic headings are retained on arrival at stopovers (here, after every fifth flight-step). Such a strategy is however somewhat inconsistent with TCSC migrants otherwise ignoring geographic headings on arrival (see the "Methods" section). Similar results were obtained when (Fig. 7d) migrants alternatively retained their headings from the first night after landing, i.e., whether departing on that night or making a longer stopover. This similarity was also found in simulations of the other long-distance migrants (Supplementary Fig. 3).

**Factors governing compass-course performance.** To diagnose factors governing compass-course performance over entire routes globally, we generalized expected performance in the normal limit (Eqs. (3) and (16)) to include parameters governing spherical-geometry effects and compass-course sensitivity (see the "Methods" section). We also estimated how seasonal constraints on migration (Table 2) limit performance. We focused on the overall best-performing loxodrome and TCSC courses, with differences between geographic and geomagnetic loxodromes indicating non-dipole (geomagnetic declination) effects[56]. For each compass course, we applied nonlinear regression and model selection, with directional precision among flight-steps as the independent variable, to fit compass-course performance among species. We predicted that the performance gain with TCSC over loxodrome

courses would depend on three (parameter-related) factors: the minimum number of flight-steps, a route-specific spherical-geometry factor (Eq. (18)) and flight-step distance (longer distances producing greater TCSC self-correction). The spherical-geometry factor increases with increasing latitude and increasingly E–W orientation (Supplementary Fig. 4).

For each compass course, the most parsimonious regression model included all relevant performance factors (i.e., flight-step distance only for TCSC), and fit performance extremely well among species ($R_{adj}^2 \geq 0.97$, see Supplementary Tables 1 and 2). Figure 8 depicts compass-course performance (solid symbols) for each species simulations and tested magnitude of flight-step precision, and also regression-estimated performance (open symbols), for geographic loxodromes (purple hexagon), geomagnetic loxodromes (orange hexagons) and TCSC courses (green circles). Species (Fig. 8a–i) are presented in increasing order of the product of the three performance gain factors. Assuming equivalently precise flight-steps among courses, TCSC courses once again consistently outperformed both loxodromes, with geomagnetic loxodromes performing less consistently than geographic loxodromes. Considering biologically relevant error scenarios, the performance gain with TCSC compared with loxodromes varied as predicted with the minimum number of flight-steps and spherical-geometry factor, as illustrated for the default error scenario with 15° compass precision in Fig. 8j, and

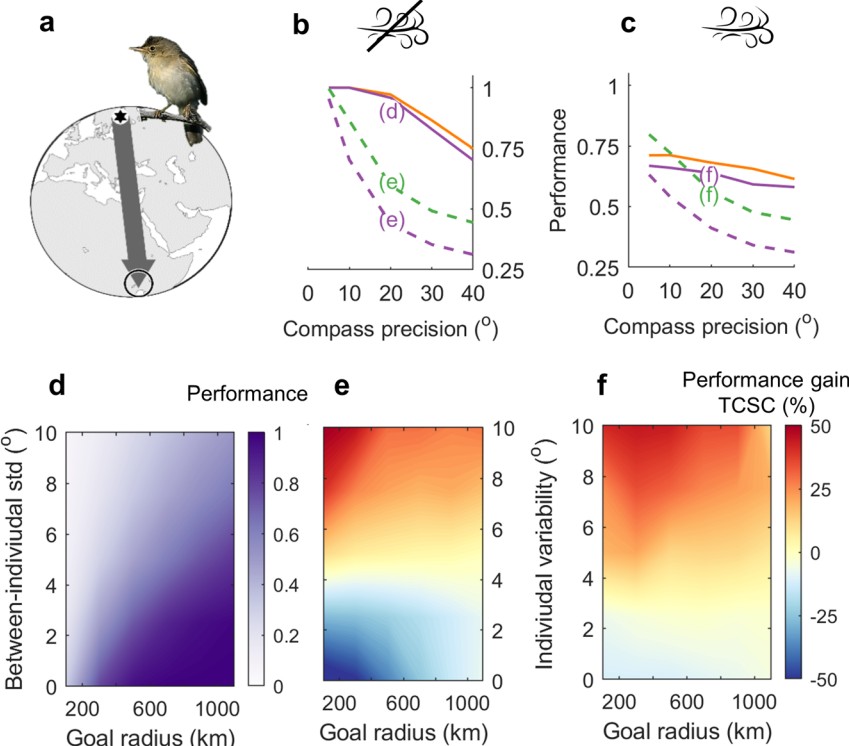

**Fig. 6 Effects of in-flight drift, individual variability and goal area on predicted marsh warbler migration. a** Marsh warbler (*Acrocephalus palustris*) migration route between breeding grounds in Finland and East Africa (grey arrow in stereographic map), with default 500-km goal radius (open circle). **b** Performance versus compass precision in the absence of drift for geomagnetic loxodromes (orange line), geographic loxodromes (purple line), cue-transferred geographic loxodromes (dashed purple line, when the star compass is unavailable on departure) and cue-transferred sun compass courses (dashed green line). **c** As in **b**, except with 20° within-flight drift. **d** Performance of geographical loxodromes (e.g., star compass) in the absence of drift as a function of goal radius and between-individual variability in inherited headings, based on 20° compass precision. **e** As in **d**, but depicting the relative performance gain with TCSC (%) over geographic loxodromes. **f** As in **e**, but with 20° within-flight drift. Photo by M. Szczepanek (https://creativecommons.org/licenses/by-sa/3.0).

for other biologically relevant error scenarios in Supplementary Fig. 5 (for scenarios with compass precision poorer than 30°, only the daytime migrant monarch butterfly favoured TCSC). The role of the gain factors in the trade-off between self-correction and cue transfer was reflected in the model-selected regression coefficients (Supplementary Table 2), with baseline performance of TCSC courses predicted to increase faster with a number of steps compared with loxodrome courses, but also "decaying" nearly twice as rapidly with decreasing flight-step precision. Error-augmentation due to the spherical-geometry factor was also three times larger along geomagnetic loxodrome courses compared with geographic loxodrome or TCSC courses, reflecting heightened sensitivity when crossing lines of declination[39,56,57].

## Discussion

Our extended formulations have facilitated a global assessment of robustness among compass courses, providing a predictive framework of naïve migratory performance and compass cue favourability among airborne migratory species and routes. Our study further highlights and quantifies three largely overlooked aspects of compass-based movements: spherical-geometry effects on course robustness, potential disadvantages of cue-transfers by naïve migrants, and that time-compensated sun compass (TCSC) courses can partially self-correct. As a result, while naïve performance regarding successful arrival is primarily constrained by directional precision and goal breadth, we further found that the relative performance gain with TCSC over courses increases with three main factors: the number of required flight-steps, flight-step distance, and a readily-derived spherical-geometry factor (Eq.

(18)), which itself increases with latitude and with more Eastward or Westward orientation (Supplementary Fig. 4). However, feasibility and favourability among compass courses remain contingent on appropriate compass mechanisms, precision, and behavioural abilities, including accommodation of confounding environmental factors.

Although it is well-recognized that compass-cue availability and compass precision can be limited at high latitudes[38,39,57], the mediating role of spherical geometry on resultant performance has been largely ignored in animal migration and navigation studies[12,36,47]. Our study emphasizes that flight-step errors at high latitudes have disproportionately large effects on compass-based movement, particularly along routes with a significant longitudinal component. Early nautical explorers overcame analogous challenges by developing maps with course headings, later improved by using transverse Mercator projections[58]. Naïve airborne migrants at high latitudes may in fact automatically mitigate such errors during early-autumn nights, with the shorter durations automatically reducing magnitudes of longitudinal errors. Our analysis also highlights how cue transfers reduce overall compass-course performance, contrasting with the advantage of combining multiple imprecise cues for experienced individuals with a map sense[18,59]. Naturally, multiple cues would still be advantageous to naïve migrants if a primary cue was unavailable (e.g., when overcast[15,46]) or unreliable (e.g., close to magnetic poles[39,57]).

The contrasting sensitivity (Fig. 3) and performance (Figs. 4 and 5) among compass courses have strong implications regarding their adaptive value to migratory populations. We

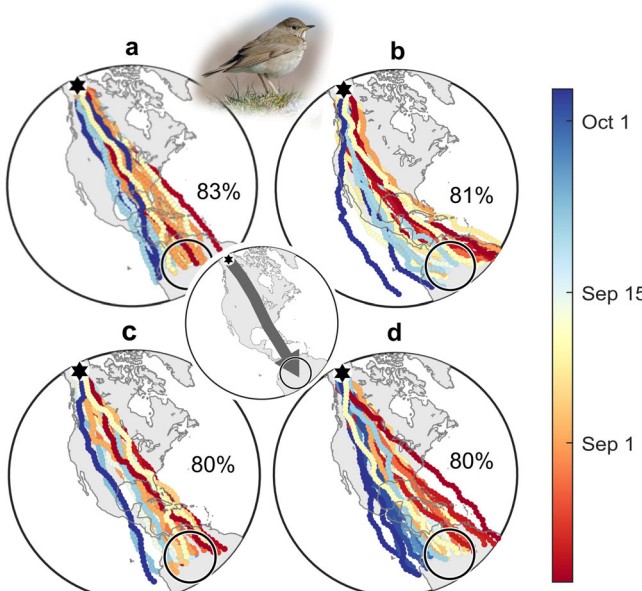

**Fig. 7 Extended sun-compass formulations illustrate flexibility in time-compensation, latitudinal and inner-clock effects.** Time-compensated sun compass (TCSC) trajectories with 20° directional precision among flight-steps, modelled after grey-cheeked thrush (*Catharus minimus*) migration (grey arrow in inset) from Yukon, Canada (black hexagons) to Columbia (open circles). Trajectories are colour-coded for initial departure date and performance (% arrival) is listed in each panel. Great circles appear as straight lines in the stereographic projection. **a** With uninterrupted nightly flights and, *sensu* Alerstam[22], adjustments in heading gauged according to (hourly) angular speed of sun-azimuth retained from the natal grounds. **b** As in **a**, but with heading-adjustments based on proximate (local) rather than natal-site speeds of sun-azimuth rotation. **c** Based on local sun-azimuth as in **b**, but including stopovers as in Fig. 4, with geographic headings retained on arrival. **d** As in **c**, but where migrants alternatively retained their headings from the first night of stopover.

propose that magnetoclinic courses are unlikely to have evolved given their general high sensitivity and poor performance along strongly direction-changing routes (for which they were envisaged[33]). Moreover, along nearly Southward routes, loxo-drome and sun compass courses perform equivalently well or better (Fig. 5d, e). Contrastingly, as an emergent "many slightly corrected wrongs" phenomenon, TCSC courses are ubiquitously more robust compared with fixed sun-compass courses, even outperforming non-transferred loxodromes for the longest-distance (willow warbler, grey-cheeked thrush and monarch butterfly) routes, and also most closely matching the known routes. This is consistent with observed flight directions of high-latitude bird migrants[56,60] most closely resembling great-circle headings (but see ref. [57]), and with primacy of twilight cues among many longer-distance migrants[30,31]. Contrastingly, for night-migratory routes at mid-latitudes, geomagnetic loxodrome and star-compass courses performed best in biological scenarios (assuming equivalent cue precision and availability). When stars are not visible on departure, loxodrome courses transferred from polarized light to a star compass performed less well (Fig. 6), even without accounting for errors in averaging cues from dusk and dawn (i.e., since this would also require a cue transfer)[24,31]. This points to a further potential advantage of nocturnal TCSC courses, which could use sunset polarized light cues without requiring a sunrise heading. Our results further support that continental-scale TCSC courses can be robust to variable scheduling of flight-steps and inner clock resetting, as well as how

headings are retained during stopover (Fig. 7). An important caveat to TCSC courses in pre-breeding (spring) migrations is that self-correction will not work for poleward movement (see also refs. [34,45]), at least without integration with additional cues.

The finding that TCSC courses are self-correcting provides a potential explanation for how naïve migrants mitigate orientation errors, but the mechanisms underlying corrective orientation by naïve migrants following displacement remain unresolved. Interpretation of experimental evidence of such corrections is often complicated by wind[48,49], polar or equatorial cue effects[39,61], and probably by resetting of inner-clocks[16,62]. Two of the three studies which tracked juvenile night-migratory birds following displacement found clear evidence of compensatory movements[48,49,63]. For eight common cuckoos (*Cuculus canorus*) tracked by satellite following a 28° longitude displacement to the East at 55°N[49], the estimated overall orientation shift relative to non-displaced (control) individuals (21°) is intriguingly close to as predicted by Eq. (4) for a TCSC (23°). However, shifts in orientation among juvenile songbirds after being displaced 16° to the West from Denmark to the Faroe Islands[48] exceeded those predicted by a self-correcting star-compass (or TCSC), as did estimated corrections from a meta-study of orientation in funnels following real and virtual displacement[42]. The mechanisms underlying all of these corrections, therefore, remain unclear[4]. To diagnose the possible involvement of celestial compass use through the displacement of naïve migrants, we recommend carefully controlling for access to celestial cues throughout the study (to assess possible resetting of inner clocks). To help distinguish between inner clock celestial and other cue effects, we further suggest displacing individuals from the same capture location to the East and West and, if possible, also clock-shifting locally captured (i.e., non-displaced) migrants.

Overall, feasibility and favourability among compass courses remain contingent upon appropriate biological cue mechanisms[2,10], relative compass precision and cue availability, much of which remains little known. Regarding cue mechanisms, it is not yet clear whether all migratory birds possess a magnetic compass or respond to sun azimuth and reset their inner clocks consistently with a TCSC[15,30,31,64]. It is even unclear whether naïve migratory bats display innate migratory directions[65]. Regarding relative compass precision and cue availability, we have avoided migration through polar regions or crossing the equator, where compass cues can become unavailable or uninterpretable[10,11,34]. Nonetheless, our model prediction that magnitudes of the compass and drift-related errors should remain below about 30° ($\kappa > 3.7$, circular length > 0.85) is supported by nightly concentrations in flight direction among radar measurements of nocturnally migrating birds[40,66]. Increased benefits from more frequent than the hourly cue maintenance used in our simulations are presumably limited by motion and cue-related effects (Supplementary Fig. 1c). Evidence of larger variability in migratory directions en route[11,12,36] likely relates in part to responses and adaptations to external environmental factors.

Actual migratory routes are naturally also contingent on and adapted to environmental factors beyond compass precision, including topography[9,28], habitat[67] and weather[29]. An important consideration is whether compass-based movement can accommodate such spatiotemporally variable factors, as well as in the Earth's geomagnetic field[68], without requiring more sophisticated (naïve navigational) abilities[4,49,69]. In the simplest case, cumulative responses to coastal and wind effects can be accommodated by an offset to a single inherited migratory headings[41,70]. More complex and detoured routes could potentially be accommodated by following sequences of innate compass headings[9,70], e.g., with shifts between headings triggered by environmental conditions, such as resource availability[67,71], or geomagnetic directional

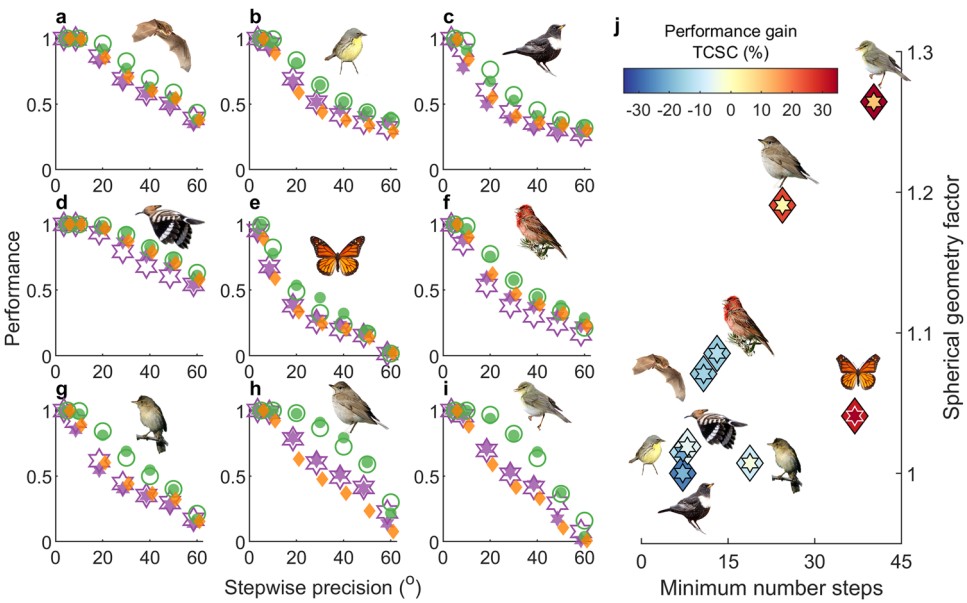

**Fig. 8 Relative compass-course performance and cue favourability predicted by distance and spherical-geometry factors. a–i** Route-optimized performance (arrival probability) vs. directional precision among flight-steps per modelled species (Table 2) for geographic loxodrome (purple hexagons), geomagnetic loxodrome (orange diamonds, right-adjusted for visibility), and TCSC courses (green circles). Species are arranged on the y-axis in increasing order of the product of the three performance gain factors (see text). Open symbols depict model-selected regression-estimated performance ($R^2_{adj} \geq$ 0.97) for geographic loxodromes (hexagons) and TCSC courses (circles), including compass-specific parameters factors governing convergence in mean heading with a number of steps, spherical-geometric effects and (for TCSC courses) flight-step distance. **j** Performance gain (%) vs. the minimum number of flight steps and the spherical geometry factor (Eq. (18)), for TCSC courses relative to geographic loxodromes (colour-coded inner hexagons) and geomagnetic loxodromes (colour-coded outer diamonds), here when additionally considering 15° compass precision including cue transfers where applicable, 15° drift, and 2.5° between-individual variability (for other error scenarios, see Supplementary Fig. 5). Photos as in Figs. 4, 6 and by **a** C. Giese, **c** P. Gomez (https://creativecommons.org/licenses/by-sa/3.0) and **d** Copyright © Albert Molenaar, via Observation.org.

signposts[10,72]. Alternatively, more advanced naïve migrant abilities beyond compass-based movement have been proposed to explain enhanced orientation correction following displacement[4], or control of naïve trans-oceanic migration routes[72]. Naïve migrants are accordingly proposed to gauge gradients in both geomagnetic intensity and inclination along their inaugural route, to either adjust (inherited) compass headings as a corrective measure[4,69], or else perform gradient-based navigation towards (inherited) geomagnetic goal signatures[73,74]. Apart from perceptive and cognitive feasibility, the efficacy of the former ability and the efficiency of the latter remains to be established, in particular given the overall N–S gradients in both geomagnetic intensity and inclination[74,75].

In conclusion, we provide a modelling framework to analyse directed compass-based movement on a spherical Earth based on spatiotemporal characteristics of geomagnetic and celestial compass cues, and incorporating precision on departure and within the flight. While predictive more than diagnostic, our results support observed diversity among migratory populations regarding compass-cue hierarchy[10,30,31], and suggest that a time-compensated sun compass can potentially lead to the highest arrival success at the wintering grounds for many naïve long-distance migrant populations. From a movement ecology perspective, our study highlights that care must be taken when assessing movement without accounting for precision in cue perception and subsequent orientation. More generally, our study illustrates how models with simple rules can potentially explain complicated patterns observed in nature, and reveal novel emergent effects with potentially profound life-history implications.

## Methods
**Calculation of flight-step headings and movement.** Terms defining flight-step movement, precision and geophysical orientation cues are listed in Table 1. Since

seasonal migration nearly ubiquitously proceeds from higher to lower latitudes, it is convenient to define headings clockwise from geographic South (counter-clockwise from geographic North for migration commencing in the Southern Hemisphere). Assuming a spherical Earth, a sequence of N migratory flight-steps with corresponding headings, $\alpha_i$, $i = 0,…, N-1$, the latitudes, $\varnothing_{i+1}$, and longitudes, $\lambda_{i+1}$, on completion of each flight-step can be calculated using the Haversine Equation[76], which we approximated by stepwise planar movement using Eqs. (1) and (2). For improved computational accuracy and to accommodate within flight-step effects, we updated simulated headings and corresponding locations hourly. A migrant's flight-step distance $R_{step} = 3.6V_a \cdot n_H / R_{Earth}$ (in radians), depends on its flight speed, $V_a$ (m/s) relative to the mean Earth radius $R_{Earth}$ (km), and flight-step hours, $n_H$. With a geomagnetic in-flight compass, expected hourly geographic headings are modulated by changes in magnetic declination, i.e., the clockwise difference between geographic and geomagnetic South[10,32].

**Formulation of compass courses.** For simplicity, we consider the case of a single inherited or imprinted heading. This can be extended to include sequences of preferred headings. Expected geographic loxodrome headings remain unchanged en route, i.e.,

$$\bar{\alpha}_i = \bar{\alpha}_0 \qquad (5)$$

Relative to geographic axes, expected geomagnetic loxodrome headings remain unchanged relative to proximate geomagnetic South, i.e., are offset by geomagnetic declination on departure (updated hourly in simulations)

$$\bar{\alpha}_i = \bar{\alpha}_0 + \delta_{m,i} \qquad (6)$$

As described and illustrated in detail by Kiepenheuer[13], the magnetoclinic compass was hypothesized to explain the prevalence of "curved" migratory bird routes, i.e., for which local geographic headings shift gradually but substantially en route. A migrant with a magnetoclinic compass adjusts its heading at each flight-step to maintain a constant transverse component, $\gamma'$, of the experienced inclination angle, $\gamma$, so that error-free headings are (see Fig. S5 in ref. [34])

$$\bar{\alpha}_i = \sin^{-1}\left(\frac{\tan\gamma_i}{\tan\gamma'}\right) = \sin^{-1}\left(\frac{\tan\gamma_i \sin\bar{\alpha}_0}{\tan\gamma_0}\right). \qquad (7)$$

In a geomagnetic dipole field, the horizontal ($B_h$) and vertical ($B_z$) field, and therefore also inclination, each depends solely on geomagnetic latitude, $\varnothing_m$: $\gamma = \tan^{-1}(B_z/B_h) = \tan^{-1}(2\sin\phi_m/\cos\phi_m) = \tan^{-1}(2\tan\phi_m)$. The projected

transverse component, therefore, becomes

$$\gamma' = \tan^{-1}\left(\frac{\tan\gamma_0}{\sin\bar{\alpha}_0}\right) = \tan^{-1}\left(\frac{2\tan\phi_{m,0}}{\sin\bar{\alpha}_0}\right),$$

which can be substituted into Eq. (7) to produce a closed formula for magnetoclinic headings in a dipole as a function of geomagnetic latitude

$$\bar{\alpha}_i(\phi_{m,i}) = \sin^{-1}\left(\frac{\sin\bar{\alpha}_0}{\tan\phi_{m,0}}\cdot\tan\phi_{m,i}\right), \qquad (8)$$

with the expected initial heading, $\bar{\alpha}_0$, and initial geomagnetic latitude, $\varnothing_{m,0}$, being constants. Equations (7) and (8) have no solution when inclination increases en route, which could occur following substantial orientation error or in strongly non-dipolar fields. We followed previous studies in allowing magnetoclinic migrants to head towards magnetic East or West until inclination decreased sufficiently[33,34,46], but also included orientation error based on the modelled compass precision.

To assess sun-compass sensitivity algebraically, and also to improve computational efficiency, we used a closed-form equation for sunset azimuth, $\theta_s$ (derived in Supplementary Note 3 and see ref. [23]),

$$\theta_s = \cos^{-1}\left(\frac{-\sin\delta_s}{\cos\phi}\right), \qquad (9)$$

where $\delta_s$ is the solar declination, which varies between $-23.4°$ and $23.4°$ with season and latitude[23]. Sunset azimuth is the positive and sunrise azimuth is the negative solution to Eq. (9) (relative to geographic N–S).

Fixed sun-compass headings represent a uniform (clockwise) offset, $\bar{\alpha}_s$ to sunrise or sunset azimuth, $\theta_{s,i}$ (calculated using Eq. (9))

$$\bar{\alpha}_i = \bar{\alpha}_s + \theta_{s,i} \qquad (10)$$

where the preferred heading on commencement of migration, $\bar{\alpha}_s = \bar{\alpha}_0 - \theta_{s,0}$, is presumed to be imprinted using an inherited geographic or geomagnetic heading[2,10,30].

With a TCSC, preferred headings relative to sun azimuth are adjusted according to the time of day. In the context of sun-compass use during migration, Alerstam and Pettersson[22] related the hourly "clock-shift" induced by crossing bands of longitude ($\Delta h = 12\,\Delta\lambda/\pi$), to a migrant's time-compensated adjustment given the rate of change (i.e., angular speed) of sun azimuth close to sunset

$$\frac{\partial\theta_s}{\partial h} \cong \frac{2\pi\sin\phi}{24}, \qquad (11)$$

resulting in a "time-compensated" offset in heading on departure ($\Delta\bar{\alpha} \cong \Delta\lambda\sin\phi$, which Eq.(4)). Equation (4) results in near-great-circle trajectories for small ranges in latitude, $\varnothing$, until inner clocks are reset. The feasibility of TCSC courses over longer distances (latitude ranges) relies on two critical but little-explored assumptions: (1) time-compensated orientation adjustments are presumed to follow the angular speed of sun azimuth (Eq. (11)) retained from the most recent clock-reset site, and (2) to negotiate unpredictable migratory schedules, migrants are presumed to retain their preferred geographic heading on arrival at extended stopovers[22].

Regarding the first assumption, time-compensated adjustments could also be influenced by proximate speeds of sun azimuth even when inner clocks are not fully reset. We, therefore, use distinct indices to keep track of "reference" flight-steps for clock-resets ($c_{ref,i}$) and time-compensated adjustments ($s_{ref,i}$). TCSC flight-step headings can then be written as

$$\bar{\alpha}_i = \begin{cases} \bar{\alpha}_{c_{ref,i}} + \left(\theta_{s,i} - \theta_{s,c_{ref,i}}\right) + \left(\lambda_i - \lambda_{c_{ref,i}}\right)\sin\phi_{s_{ref,i}}, & i \neq c_{ref,i} \;(12a) \\ \alpha_{i-1}, & i = c_{ref,i} \;(12b) \end{cases},$$

where $\theta_{s,i}$ represents the sunset azimuth on departures, $c_{ref,i}$ specifies the most recent clock-reset site (during which geographic headings are also retained, i.e., $\bar{\alpha}_i = \bar{\alpha}_0$), and $s_{ref,i}$ specifies the site defining the migrant's temporal (hourly) rate of "time-compensated" adjustments (Eq. (11)). For TCSC courses as conceived by Alerstam and Pettersson[22], reference rates of adjustment to sun azimuth are reset in tandem during stopovers, i.e., $s_{ref,i} = c_{ref,i}$, but we also considered a proximately gauged TCSC, where migrants gauge their adjustments to currently experienced speed of sun azimuth, i.e., $s_{ref,i} = i$.

Regarding the second assumption, retaining geographic headings on arrival at stopovers is not consistent with ignoring geographic headings between consecutive nightly flight-steps, and may be difficult to achieve while landing. We, therefore, examined a more parsimonious alternative (Fig. 7d, Supplementary Fig. 3) where migrants retain their (usual) TCSC heading from the first night of stopovers, i.e., as if they would have departed on the first night. This alternative also simplifies Eq. (12) to

$$\bar{\alpha}_i = \bar{\alpha}_{c_{ref,i}} + \left(\theta_{s,(t_{i-1}+1)} - \theta_{s,t_{i-1}}\right) + \left(\lambda_i - \lambda_{c_{ref,i}}\right)\sin\phi_{s_{ref,i}} \qquad (12c)$$

where the index $t_{i-1}$ here represents the departure date from the previous flight.

**Sensitivity of compass-course headings.** Sensitivity was assessed by the marginal change in expected heading from previous (imprecise) headings, $\partial\bar{\alpha}_i/\partial\alpha_{i-1}$. When this is positive, small errors in headings will perpetuate, and therefore expected errors in migratory trajectories will grow iteratively. Conversely, negative sensitivity implies self-correction between successive flight-steps. Geographic and geomagnetic loxodromes are per definition constant relative to their respective axes so have "zero" sensitivity, as long as cue-detection errors are stochastically independent.

For magnetoclinic compass courses in a dipole field, sensitivity can be calculated by differentiating Eq. (8) with respect to previous headings:

$$\frac{d\bar{\alpha}_i}{d\alpha_{i-1}} = \frac{\sin\bar{\alpha}_0}{\tan\phi_{m,0}}\cdot\frac{1}{\cos\bar{\alpha}_i\cos^2\phi_{m,i}}\frac{\partial\phi_{m,i}}{\partial\alpha_{i-1}} = \frac{R_{step}\,\sin\alpha_{i-1}\sin\bar{\alpha}_0}{\cos\bar{\alpha}_i\cos^2\phi_{m,i}\,\tan\phi_{m,0}} \qquad (13)$$

All three terms in the denominator indicate, as illustrated in Fig. 3b, that magnetoclinic courses become unstably sensitive at both high and low latitudes, and any heading with a significantly East–West component.

Sensitivity of fixed sun compass headings is non-zero due to sun azimuth dependence on location (Eq. (9)):

$$\frac{d\bar{\alpha}_i}{d\alpha_{i-1}} = \frac{\sin\delta_{s,i}}{\sin\theta_{s,i}}\cdot\frac{\sin\phi_i}{\cos^2\phi_i}\frac{\partial\phi_i}{\partial\alpha_{i-1}} = \frac{\sin\delta_{s,i}}{\sin\theta_{s,i}}\cdot\frac{R_{step}\,\sin\phi_i\,\sin\alpha_{i-1}}{\cos^2\phi_i}$$
$$= R_{step}\cdot\sin\alpha_{i-1}\frac{\tan\phi_i}{\tan\theta_{s,i}} \qquad (14)$$

The sine factor on the right-hand side in Eq. (14) causes the sign of $\partial\bar{\alpha}_i/\partial\alpha_{i-1}$ to be opposite for East to West or West to East headings, and $\tan\theta_s$ also change sign at the fall equinox (due to solar declination changing sign). The azimuth term in the denominator indicates heightened sensitivity closer to the summer or winter equinox and at high latitudes, and, conversely, heightened robustness to errors closer to the spring or autumnal equinox (since $\tan\theta_{s,0} \to \pm\infty$). This seasonal and directional asymmetry is illustrated in Fig. 3c, e.

TCSC courses (Eq. (12)) involve up to three sensitivity terms, due to dependencies on sun azimuth, longitude and latitude:

$$\frac{d\bar{\alpha}_i}{d\alpha_{i-1}} = R_{step}\cdot\sin\alpha_{i-1}\frac{\tan\phi_i}{\tan\theta_{s,i}} + \frac{d\lambda_i}{d\alpha_{i-1}}\sin\phi_{ref,i} + \left(\lambda_i - \lambda_{c_{ref,i}}\right)\frac{d\sin\phi_{s_{ref,i}}}{d\alpha_{i-1}}$$
$$= \begin{cases} R_{step}\cdot\left[\sin\alpha_{i-1}\frac{\tan\phi_i}{\tan\theta_{s,i}} - \frac{\cos\alpha_{i-1}\sin\phi_{ref,i}}{\cos\phi_{i-1}}\right], & \text{classic (15a)} \\ R_{step}\left[\sin\alpha_{i-1}\frac{\tan\phi_i}{\tan\theta_{s,i}} - \frac{\cos\alpha_{i-1}\sin\phi_{ref,i}}{\cos\phi_{i-1}} + \left(\lambda_i - \lambda_{c_{ref,i}}\right)\sin\alpha_{i-1}\cos\phi_i\right], & \text{proximate (15b).} \end{cases}$$

The first square-bracketed terms in Eqs. (15a, b) are identical to the fixed sun compass (Eq. (14)), reflecting seasonal and latitudinal dependence in sun-azimuth. For headings with a Southward component ($\alpha_0 < 90°$), the second bracketed terms are always negative, i.e., sensitivity-reducing, resulting in a broad range in latitude and headings with self-correcting headings (Fig. 3c–f). The third bracketed term in Eq. (15b) (for a proximately-gauged TCSC) is also negative, and in fact increasingly so until clocks are reset, but remains small in magnitude compared to the second term.

**Spatiotemporal migration model.** We wrote a model in MATLAB to simulate and assess the feasibility and robustness of each compass course to spatiotemporal effects on a global scale, based on our compass course formulations (Eqs. (5)–(12)). For the species simulations, we also incorporated spatiotemporally dynamic geomagnetic data (MATLAB 2020b package igrf)[51], assuming a default season, fall 2000. For the generic migrant simulations, we assumed a geomagnetic dipole Earth, i.e., ignored variation in magnetic declination. Sunset azimuth was computed using Eq. (9) (this was, e.g., two orders of magnitude faster than the routine further requiring time of day and longitude used in ref. [34]). In all cases 10,000 individuals were simulated, until they either arrived in goal areas, passed 1000 km South of the goal latitude or exceeded the maximum number of steps, $N_{max}$. To avoid migrants overshooting narrow goal areas within a single flight-step, we assumed they could identify goal areas in flight (checked once per decile of flight-step durations; Table 2).

For the species simulations, optimal inherited headings were determined using the MATLAB nonlinear solver *fminbnd*, among candidate initial headings clockwise from East ($-90°$ clockwise from S) to West ($90°$). For sun compass courses, which can potentially begin with Northward headings[34,64], we tested initial headings between NE ($-145°$) and NW ($145°$). Error scenarios were assessed for both 5°–60° directional precision among flight-steps, and biologically relevant scenarios with 5°–40° compass precision in 5° intervals (assuming equivalent precision in cue detection, transfers and maintenance), and both in the absence of and including 15° within-flight drift. To result in 15° expected drift per flight step, hourly concentration in drift was adjusted as an autocorrelated process with lag 1. We further assumed a default between-individual variability of 2.5° ($\kappa = 525$, circular length = 0.999). For the marsh warbler uncertainty analysis (Fig. 6), we varied this between 0° and 10° ($\kappa = 33$, circular length = 0.985) in 1° interval, and also tested 20° within-flight drift.

For the generic migrants, we simulated migration between 65°N–0°N and between 45°N–25°N to a goal with a radius of 500 km, for biologically relevant scenarios with 0°–60° compass precision in 1° interval, and both in the absence of and including 15° within-flight drift. To obtain all possible routes between these latitudes (Fig. 5a), we varied initial (inherited) headings in 0.5° intervals.

All relevant model parameters are listed in Table 2. These were chosen to match known studies and migration patterns. In some cases (e.g., marsh warbler and monarch butterfly), goal areas reflect plausible destinations from which migrants presumably use other cues to home or pin-point to even narrower known winter or passage areas[2,3]. When flight-step distances and stopover durations were less known or certain, these were chosen to ensure modelled migration was consistent with known departure and arrival dates.

Generic migrants departed on September $15 \pm 5$ (mean and standard deviation), in 8-h flight-steps at speeds of $12.5\,\mathrm{m\,s^{-1}}$. Sequences of 5 consecutive flight-steps were interspersed by stopovers of $5 \pm 2$ days.

**Formulating migratory performance**. Performance (arrival probability) of independent stepwise movement on a plane to a (circular) goal area of radius $R_{goal}$ will approximate a cumulative normal distribution (erf function), modulated by the expected number of steps and the angular breadth to the goal area, which for long-distance routes is $\beta = \tan^{-1}\left(R_{goal}/R_{mig}\right) \cong R_{goal}/R_{mig}$. Assuming uniform population headings and a sufficiently large number of flight-steps with directional precision $\sigma_{step}$, a first approximation of performance is

$$\hat{p}_{\beta,\hat{N}} \approx p\left(\left|\left(\frac{1}{\hat{N}}\sum_{i=1}^{\hat{N}}\alpha_i\right) - \bar{\alpha}\right| \le \beta\right) \approx erf\left(\frac{\beta}{\sqrt{2}\sigma_{step}/\sqrt{\hat{N}}}\right), \quad (16)$$

where the expected number of steps, $\hat{N}$, scales with the minimum (error-free) number of steps,

$N_0$, multiplied by a ratio of Bessel functions (Supplementary Note 2). From Eq. (16) we see that within the planar and high-precision limits, performance will increase with $\beta_{adj} = \beta\sqrt{N_0}$, which is the length-adjusted goal breadth (Eq. (3)).

In the equation for flight-step longitude (Eq. (2)), the secant factor (cosine of latitude in the denominator) reflects the poleward convergence of longitudinal meridians. This means that for any compass course, orientation errors at higher latitudes will exert a greater influence on overall longitudinal error:

$$\frac{d\lambda_i}{d\alpha_{i-1}} \cong -\frac{R_{step}\cos\alpha_{i-1}}{\cos\phi_{i-1}}. \quad (17)$$

Aggregated across entire migration routes, the effective longitudinal error will scale approximately as in a Mercator projection[58]:

$$L = \frac{1}{(\phi_0 - \phi_A)}\int_{\phi_A}^{\phi_0}\frac{d\phi}{\cos\phi} = \frac{1}{(\phi_0 - \phi_A)}\ln\left(\frac{\tan(\phi_0/2 + \frac{\pi}{4})}{\tan(\phi_A/2 + \frac{\pi}{4})}\right)$$

where $\varnothing_0$ and $\varnothing_A$ are the initial (natal) and arrival latitude, respectively. To include latitudinal contribution to error, we modulated the multiplicative factor $L$ according to route-mean orientation, $\bar{\alpha}$,

$$G = \sqrt{(L\sin\bar{\alpha})^2 + \cos^2\bar{\alpha}}. \quad (18)$$

For TCSC courses $\bar{\alpha}$ was estimated as the average between the initial and final great circle bearings between the natal and goal locations. The spherical-geometry factor, $G$, is largest for purely Eastward or Westward headings ($G = L > 1$) and nonexistent for North–South headings ($G = 1$, reflecting no longitude bands being crossed). We expected this factor to affect compass courses differentially according to their error-accumulating or self-correcting nature.

We further modified the effective goal-area breadth to account for a (geographically) circular goal area on the sphere, i.e., effectively modulating the longitudinal component of the goal-area breadth at the arrival latitude, $\varnothing_A$:

$$\beta_A = \beta\sqrt{\sin^2\bar{\alpha} + \left(\cos\bar{\alpha}/\cos\phi_A\right)^2}. \quad (19)$$

To account for differential sensitivity among compass-courses, we generalized the normal many-wrongs relation between performance and number of steps, $1/\hat{N}^\eta$, from $\eta = 0.5$ (Eqs. (3) and (16)) to

$$\eta\left(\sigma_{step}|s,b\right) = (0.5 + b)e^{-s\sigma_{step}^2}, \quad (20)$$

where $b < 0$ reflects iterative augmentation of errors and $b > 0$ self-correction, and $s$ represents a modulating exponential damping factor, consistent with the limiting circular-uniform case (as $\kappa \to 0$, i.e., $\sigma_{step} \to \infty$), where no (timely) convergence of heading is expected with an increasing number of steps.

In assessing performance, we also accounted for seasonal migration constraints via a population-specific maximum number of steps, $N_{max}$ (Table 2; this became significant for the longest-distance simulations with large expected errors, i.e., small $\kappa_{step} = 1/\sigma_{step}^2$). The probability of having arrived at the goal latitude can be estimated using the Central Limit Theorem:

$$p_{\phi,N_{max}} \cong \frac{1}{2}\left[1 - erf\left(\left(\frac{N_0}{N_{max}} - \frac{I_1\left(\kappa_{step}\right)}{I_0\left(\kappa_{step}\right)}\right)\cdot\frac{\cos\bar{\alpha}}{\sigma_C\sqrt{2}}\right)\right], \quad (21)$$

where $I_j$ is the modified Bessel function of the first kind and order $j$[53], and $\sigma_C$ (the standard deviation in the latitudinal component of flight-step distance) can be

calculated using Bessel functions together with known properties of sums of cosines[53,77] (Supplementary Note 2).

**Regression-estimated performance**. We fit the parameters in the spherical-geometry factor (Eq. (18)) and many-wrongs effect (Eq. (20)) according to expected performance, estimated as the product of sufficiently timely migration (Eq. (21)) and sufficiently precise migration, now generalized from Eq. (16), i.e.,

$$p_{\beta,\hat{N}} \cong erf\left(\frac{\beta_A}{G^g\sqrt{2\left(\sigma_{ind}^2 + \sigma_{step}/\hat{N}^n\right)}}\right), \quad (22)$$

This resulted in up to four fitted parameters for each compass course

i. an exponent, $g$, to the spherical-geometry factor (Eq. (19)), i.e., $G^g$, reflecting how growth or self-correction in errors between steps further augments or reduces this factor,

ii. a baseline offset, $b_0$, to the "normal" exponent $\eta = 0.5$, which mediates the relation between the number of steps and performance (Eq. (20)),

iii. an exponent $s$ reflecting how decreasing precision among flight-steps dampens the many-wrongs convergence (Eq. (20)),

iv. for TCSC courses, a modulation, $\rho$, to the offset, $b_0$, quantifying the extent to which self-correction increases with increased flight-step distance $R_{step}$, i.e., $b = b_0 R'^{\rho}_{step}$ in Eq. (20), where $R'_{step}$ is the flight-step distance scaled by its median value among species.

Parameters were fit using MATLAB routine *fitnlm* based on compass course performance among species and seven error scenarios (5°, 10°, 20°, 30°, 40°, 50°, and 60° directional precision among flight-steps), for all combinations (including or excluding the four parameters). The most parsimonious combination of parameters was selected using MATLAB routine *aicbic*, based on the AICc, the Akaike information criterion corrected for small sample size[57]. Null values for the spherical-geometry parameter were set to $g = 1$, and for the parameters governing convergence of route-mean headings $b_0 = 0$, $s = 0$, and, for TCSC courses, $\rho = 0$ (for loxodrome courses, $\rho = 0$ by default, i.e., was not fitted).

**Statistics and reproducibility**. Our simulation results, regression fitting and AICc-model selection are reproducible using the MATLAB scripts (see the section "Code availability").

**Reporting summary**. Further information on research design is available in the Nature Research Reporting Summary linked to this article.

## Data availability

All data required to reproduce the Results and Figures is contained in the main manuscript, Supplementary Information and model (see the section "Code availability").

## Code availability

The code to simulate the model and reproduce all the figures is available in github repository https://github.com/jdmclaren/compass_course_model.

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

## Acknowledgements

The authors would like to thank H. Mouritsen and M. Winklhofer for fruitful discussions, and R. Holland, K. Thorup and an anonymous reviewer for important suggestions to improve the manuscript's scope and interpretation. Funding was provided by German Research Foundation SFB 1372 "Magnetoreception and navigation in vertebrates" (project number 395940726; INST 184/205-1) to B.B. and H.S., employing J.D.M.

## Author contributions
All authors worked to conceive the study. J.D.M. formulated and coded the models, analysed the results, and wrote the draft and revised manuscript. H.S. and B.B. supervised the study and project, and contributed to manuscript revisions. All authors read and commented on the manuscript and gave final approval for publication. All authors agreed to be held accountable for the work performed therein.

## Funding

## Competing interests
The authors declare no competing interests.
