## [Peer Review File · Communications Biology]

Reviewers' comments:

Reviewer #1 (Remarks to the Author):

The MS presented tackles the intriguing question of what compass mechanisms control migration in naïve migrants (clock and compass mechanism). There have been previous attempts to model expected trajectories under different hypothesised mechanisms and this builds on those and is well thought through. Overall therefore, this is a worthwhile endeavour and I think the MS is certainly of sufficient interest to warrant publication – indeed I suspect it will be well cited. I especially like the idea that 'simple' compass mechanisms might actually explain apparent correction for displacement in migrant birds.

Unfortunately, I did find the present MS extremely difficult to read throughout (to be honest, there were almost no sentences that I understood the first time of reading) and I don't think this is just a result of the technical nature of the content but rather that the authors need to really think carefully about how to convey the purpose and method of their MS. Further, the paper is formatted introduction – results – discussion – methods, but the authors expect the reader to go from the introduction to the methods and then back to the results (and then to and from tables that aren't really explained in the main text). In this intro-results format, a much clearer and more detailed proposal of the models/simulations needs to be given at the end of the introduction and the results need to be written to reflect this. In my view, therefore, the authors need to think carefully about how to explain the paper much more clearly, and in a manner suitable for the journal format. I apologise for being so vague in a criticism, and really do think that the content is rather interesting, but it is just so hard to read! I've tried to give some examples below of where clarity could be improved, but to me the paper needs a fair amount of work to make it accessible above and beyond the examples below.

Regarding the authors' simulation approach: If I understand correctly, the models are selected based on the proportion of a simulated population to successfully arrive at the destination. I wonder whether the authors might consider other factors that, in nature, could exert selective pressures on migratory routes (e.g. that survival over large expanses of water might be poor, or that mountain ranges might be obstructive). I don't propose the authors change their modelling approach, because of course these local phenomena make mechanisms likely less generalisable. But some of these obstructions are probably a large and permanent feature in the evolutionary history of many related species. For the Euro-African flyways, for example, avoiding the Mediterranean to the west or east is clearly encoded into naïve compass orientation of many species and so it might be interesting to note which of the proposed models might provide a trajectory that misses the Mediterranean best. Related to this, a subtext of the paper appears to be that birds ought not be able to inherit complex map-like information. It might be nice to see the authors discuss that a little, also, since (while unlikely perhaps) dog-legging is a real alternative to explain how birds achieve effective naïve navigation to surprising precise destinations.

Some small comments on other parts of the paper:

Abstract

Perhaps the authors might re-consider some of the more technical phrasing like "narrow-front", and try to explain in simple words how the simulations might enhance our understanding of navigation (e.g. different cues, different ways of using the cues and the physical possibility of them to deliver population-wide phenomena observed in nature).

Introduction

Page 3

L6: "innate or inherited": do these differ?

L6: While I do think I understand what the authors mean, I find phrases like "stepwise relative to proximate geophysical compass cues en route" very difficult. Throughout, the use of the word stepwise is slightly confusing (this is not because it is necessarily wrong, but it would be useful for the authors to be a clearer what they mean).

L11-12: the star compass model most favoured does not require time compensation. Authors discuss the role of the time-compensated model in the discussion but might be improved by being more explicit in both places (intro/discussion).

L20-24: an example of a sentence that is difficult to understand: "1) geographic loxodromes, following constant headings relative to geographic South or North, which is identifiable either by a primary star compass^{7,9} or else by averaging (more reliably available) maximum bands of polarized light at sunrise and sunset¹⁰⁻¹² "

The clarity might be enhanced if the geometry of the expected trajectory (constant heading relative to geographic north) is separated slightly from the mechanisms that might provide this. Throughout the introduction, the authors attempt to give all the information at the same time which makes it really tricky to follow!

Page 4

L1-5. I'm not sure that following a constant heading relative to sunrise or sunset azimuth is really menotactic, in that by paying attention to the sun's azimuth only at dusk and dawn it is quasi-time compensated. Perhaps useful here would be to consider time-limited compass information and whether that might be sufficient given an appropriate heading indicator (for semantic distinction see Guilford and Taylor 2014).

Page 25

L10-16: I think this is a very nice finding of the paper, but it would be lovely to have the authors write out in words in the main text their TSCS self-correction model in the introduction or here (and other key models, perhaps choose the ones that are discussed the most) rather than simply sending the reader to the equation and methods.

Reviewer #2 (Remarks to the Author):

This is a most interesting and important manuscript. Having been involved in quite some modelling/simulation work in the past, I very much welcome the authors' efforts to extend previous simplified migration modelling/simulation in the plane to the sphere, to using compasses associated with the rotating earth and to cue-transfers. It also means that I will necessarily be comparing with work of my own and I apologise in advance for referring more to my own work than I like or what it maybe deserves. Also, I have had some trouble in getting access to the relevant literature because of current corona restrictions.

The work allows for a revised perspective on the consequences of the use of different compasses. However, I am less convinced about the authors inference based on their modelling/simulations. The uncertainties in the underlying model parameters are large complicating making firm conclusions about the compasses used. Perhaps more important, congruence between simulations and observed patterns can, at the conceptual level, mainly lead to conclusions, such as "a simple mechanism can lead to the observed patterns", that can then be sought further validated/tested. Lastly, I do not believe that the authors present evidence that compensation observed in naïve migrants can be

explained by enhanced migration performance as the authors seem to claim. I should add that the manuscript includes extensive material that cannot be thoroughly reviewed in the short time allowed by the journal. Thus, I have only critically examined what I believe are key (biological) issues and have assumed that the formulas in the method section are correct/meaningful.

An important theme is the potential correction/apparent compensation caused by the use of a time-compensated sun compass without resetting of the internal clock (originally called pseudo-navigation by Rabøl 1998) or with compass course transfer in combination with resetting of the clock. It is not really a new idea – for instance it was discussed extensively in Thorup and Rabøl (2007). It is of course of considerable interest to model/simulate the effects along migration routes. But it is not made clear to what degree the “new” conclusions result from use of different parameters or inclusion of additional components. Indeed some previous work exists and I was surprised to not find it more directly cited and compared to the presented work (for example simulation of marsh warbler migration in Thorup et al. 2007). Also, previous literature on a similar subject has used slightly different annotation. You might of course have good reasons for choosing a different one but providing some reference to previous work would make comparison more easy. Variation in migration direction is in most work incorporated as the concentration of a von Mises distribution but here as effective step-wise error. It would help the reader (and the reviewer) to know how these two compare. In the present work, effective standard errors of 2.5deg in between-individual headings are used but how does that compare to an r -among of 0.98 where 95% of the angles fall inside ± 22 deg of the mean angle? Did the authors include less variation among individuals than in Thorup et al 2007? (I do not have access to the literature to find out myself – but I also think the authors should provide this information).

Re the suggestion/conclusion that the self-correcting sun compass can explain observed responses to displacements, they make a case only for the cuckoo displacements – obviously, the longitudinal displacement of 14deg in the Faroes displacement (Thorup et al 2011) does not render clock-shift a likely explanation of the much larger changes observed (68deg). “For the perhaps clearest and most convincing case for naïve self-correction, by GPS-tagged Eurasian Cuckoos (*Cuculus canorus*) following a 28deg longitude displacement at 55degN 20, the estimated shift in headings compared with 15 non-displaced “control” individuals (21deg) is intriguingly close to as predicted (23deg) using Equation (21)”. However, with a 28deg longitudinal change the predicted clock-shift change is 28deg – the 23deg is the change expected if birds fly themselves around sunset/sunrise without resetting; the 21deg though is still not significantly different from this. But more important, accepting the time-compensated sun compass with inner clocks being updated and last headings maintained during stopover, the routes should curve southward and the with the observed angle after 500 km the birds would not reach the normal migration route as observed in the study (where the change in direction was on average kept throughout).

The main argument against ‘pseudo-navigation’ has been that the birds had ample time to reset their internal clock allowing birds to recalibrate whatever their preferred direction is and making it independent of the days before. In principle, a bird could arrive after displacement at a new location, note landmarks for their preferred (previous-location) time-compensated sun compass direction, and then transfer this new direction to their new preferred sun compass direction when they take off days after (which is different from a bird continuing in a direction it has flown on its own). I find this quite odd and without much evidence to support it. I believe there is little evidence to suggest that migrants in general follow great circle routes apart from in very specific settings (such as in Alerstam et al 2001 without resetting presumed or Thorup (1998) Vagrancy of Yellow-browed Warbler *Phylloscopus inornatus* and Pallas's Warbler *Ph. proregulus* in north-west Europe: Misorientation on great circles? *Ringing & Migration* 19:7-12). And, I have difficulties envisioning how it should work with the transfer of daytime cues for the translocated migrants where the clock is likely to be maintained more by light-dark cycles than actual sun azimuth changes.

Specific comments (random):

P215: "airborne compass courses" – what is that?

P217: "previously overlooked spherical-geometry" – were they overlooked or have people just not considered navigation on a sphere? Perhaps just delete previously overlooked.

P2111: "increasing goal-area" – can an area increase?

P2112-14: "Our results can explain enhanced naïve migrant performance, observed diversity in compass-cue hierarchies, and sun-compass orientation being key to many long-distance inaugural migrations" – there is something odd with the sentence structure combining "can explain" with "being key". Furthermore, as outlined above I find the conclusion too far-reaching.

P319: "Birds" – too general; it has by far not been shown across species.

P315onwards: there is little distinction between what birds have been shown to be capable of (typically in cages) and what they actually do when migrating (ie. free-flying).

P416: you might want to mention that this conclusion was questioned – see the debate Mouritsen (1998), Thorup et al (2000), Mouritsen (2000) in Animal Behaviour.

P5results: I found it very difficult to understand the results without extensive reference to the methods section to find out what has been done – perhaps provide some more information either in the results section or the last part of introduction to assist interpretation.

Table 2: why use a different goal radius than Thorup and Rabøl 2007 which you cite? The 330 km step length also deviates substantially from previous estimates. Mouritsen (1998) estimated r-step as 0.665 for ring recoveries 100-150 km distant – how does that compare to the 20deg effective step error and 300-360 km step-wise distance you consider?

P2312: "first global assessment of factors governing robustness" - Apart from the unnecessary "first" you don't seem to assess the factors governing robustness but rather assess whether the factors involved govern robustness.

P2516-7: how can refs 9+29 support that something proposed in ref 45 is "now generally regarded as not supported" when ref 45 is published after 9+29?

P30eq12-13: N_0 defined in (12) does not seem to be the one inserted (13)

P37123: what is the concentration of a von Mises distribution with 2.5deg effective standard error?

Kasper Thorup

We were extremely pleased to read that the two reviewers found our manuscript most interesting and important. We very much appreciated their discerning and constructive comments. We have thoroughly revised our manuscript accordingly, to improve its accessibility, better relate to previous work and clarify its broader context in migratory navigation and orientation.

Yours sincerely,

James McLaren and co-authors

Note: for clarity, we have indented and written our responses for each comment in blue font, and quote relevant revised text in red font.

Referee expertise:

Referee #1: Bird navigation, sun compass

Referee #2: Bird navigation/migration, juvenile bird orientation mechanisms

Reviewers' comments:

Reviewer #1 (Remarks to the Author):

R1.1 The MS presented tackles the intriguing question of what compass mechanisms control migration in naïve migrants (clock and compass mechanism). There have been previous attempts to model expected trajectories under different hypothesised mechanisms and this builds on those and is well thought through. Overall therefore, this is a worthwhile endeavour and I think the MS is certainly of sufficient interest to warrant publication – indeed I suspect it will be well cited. I especially like the idea that ‘simple’ compass mechanisms might actually explain apparent correction for displacement in migrant birds.

We are pleased with your enthusiasm for the overall research question and approach to consider ‘simple’ compass mechanisms.

R1.2 Unfortunately, I did find the present MS extremely difficult to read throughout (to be honest, there were almost no sentences that I understood the first time of reading) and I don't think this is just a result of the technical nature of the content but rather that the authors need to really think carefully about how to convey the purpose and method of their MS. Further, the paper is formatted introduction – results – discussion – methods, but the authors expect the reader to go from the introduction to the methods and then back to the results (and then to and from tables that aren't really explained in the main text). In this intro-results format, a much clearer and more detailed proposal of the models/simulations needs to be given at the end of the introduction and the results need to be written to reflect this. In my view, therefore, the authors need to think carefully about how to explain the paper much more clearly, and in a manner suitable for the journal format. I apologise for being so vague in a criticism, and really do think that the content is rather interesting, but it is just so hard to read! I've tried to give some examples below of where clarity could be improved, but to me the paper needs a fair amount of work to make it accessible above and beyond the examples below.

We acknowledge that our manuscript was presented in an overly technical and condensed manner, and was not well enough organized considering the journal format.

Based on your comments, as detailed below, we have revised the Abstract and Introduction to more clearly state our context (e.g., comments R1.3, R1.6) aims (e.g., comment R1.4), and to better describe the models and approaches used (e.g., R1.3). For more direct comparison, we additionally provide four key equations in the Results (e.g., R1.10).

R1.3 Regarding the authors' simulation approach: If I understand correctly, the models are selected based on the proportion of a simulated population to successfully arrive at the destination. I wonder whether the authors might consider other factors that, in nature, could exert selective pressures on migratory routes (e.g. that survival over large expanses of water might be poor, or that mountain ranges might be obstructive). I don't propose the authors change their modelling approach, because of course these local phenomena make mechanisms likely less generalisable. But some of these obstructions are probably a large and permanent feature in the evolutionary history of many related species. For the Euro-African flyways, for example, avoiding the Mediterranean to the west or east is clearly encoded into naïve compass orientation of many species and so it might be interesting to note which of the proposed models might provide a trajectory that misses the Mediterranean best.

Related to this, a subtext of the paper appears to be that birds ought not be able to inherit complex map-like information. It might be nice to see the authors discuss that a little, also, since (while unlikely perhaps) dog-legging is a real alternative to explain how birds achieve effective naïve navigation to surprising precise destinations.

You are correct that we assess migratory performance by proportional arrival. We now mention this more explicitly in the Abstract (P2 L5) and Intro (P7 L5). We also assess courses by their robustness to error (P7 L6-7), accounting for compass precision and directional precision among flight steps, as we now state more explicitly at the end of the Introduction (P7 L16-21).

We of course agree that many other factors can shape migratory routes, as we now mention in the Introduction (P6 L5-8). We decided not to incorporate these additional environmental drivers in our model because (as correctly stated by the reviewer) they would make our conclusions less generalizable. We instead have now expanded our treatment of this in the Discussion (P33 L3-13), including the potential for naïve migrants to follow inherited directional signposts, i.e., leading to “dogleg” routes (P33 L10-12). In fact, we have addressed the feasibility of doglegs (*Zugknicks*) in a separate, to be submitted study (using an evolutionary approach with spatially heterogeneous populations), but decided that it was better to first present the more fundamental issues presented here.

Regarding which courses can best circumnavigate the Mediterranean, our Supplementary Fig. 2 (similarly to Fig. 3 in Muheim et al *Mvt Ecol* 2018) illustrates that autumnal TCSC and loxodrome courses follow similar – and similarly robust – paths along the SW and SE European-African flyways, but cannot additionally reach Central Africa without a *Zugknick*.

Some small comments on other parts of the paper:

Abstract

R1.4 Perhaps the authors might re-consider some of the more technical phrasing like “narrow-front”, and try to explain in simple words how the simulations might enhance our understanding of navigation (e.g. different cues, different ways of using the cues and the physical possibility of them to deliver population-wide phenomena observed in nature).

Thank you for this good and general suggestion.

Throughout the manuscript, we have revised to avoid where possible “condensed” terminology such as “narrow-front”, now replaced with “reliably reach remote destinations” (P2 L3). See also revision of the term “stepwise” below.

In the Abstract, we now use separate sentences to first describe inherited compass headings in one sentence (P2 L1-2), then the open question of how migrants reach remote destinations, in “daily or nightly flight-steps” (P2 L2-4). We then more explicitly describe our goal (P2 L4-5) “To identify key factors predicting naïve migratory performance (successful arrival) and compass-cue hierarchy, ...”.

We have also revised the Introduction to better describe our motivation in broader context, e.g., by stating (P3 L12-14) “However, the extent to which such compass courses (often termed clock and compass migration in the literature) can reliably reproduce observed migration patterns remains uncertain.”. We have also added a paragraph (P6 L16-24) describing possible benefits and mechanisms of migratory self-correction.

Also, we now conclude in the Discussion (P33 L 20-22) how “our study illustrates how models with simple rules can potentially explain complicated patterns observed in nature”.

Introduction

Page 3

R1.5 L6: “innate or inherited”: do these differ?

Indeed - this phrase was not well chosen (we wanted to consider that initial migratory headings could be learned or imprinted using inherited information from another compass system, e.g., as we have modelled with the sun compass).

We now therefore replaced the phrase “innate or inherited migratory headings” with “inherited migratory directions” (P3 L10-11).

R1.6 L6: While I do think I understand what the authors mean, I find phrases like “stepwise relative to proximate geophysical compass cues en route” very difficult. Throughout, the use of the word stepwise is slightly confusing (this is not because it is necessarily wrong, but it would be useful for the authors to be a clearer what they mean).

We agree that this is an issue, and have revised to reduce our reliance on technical terms and, especially, to avoid stringing these together. We also agree that using

“stepwise” to describe direction changes or precision is ambiguous (particularly since we subdivide flight-steps hourly). We therefore now avoid this and refer specifically to “directional precision among daily or nightly flight-steps” (P7 L15-17) and to “compass precision” (P2 L7, P7 L17-20). We now use the word stepwise sparingly according to common English usage, i.e., “a sequence of steps”, e.g., in the caption of Table 2 “...terms describing stepwise movement” (P9 L1).

For the specific phrase you cite, we rewrote the sentence as three separate sentences (P3 L6-12). The first [1] describes the independent migration of many airborne migrants (P3 L6-8), the second [2] that long-distance migrants typically fly in a sequence of daily or nightly flight-steps (P3 L8-10), and the third [3] that they are thought to accomplish this using various compasses (P3 L10-12).

“[1] However, many naïve airborne migrants complete their journeys to population-specific remote destinations (hereafter, goal areas) independently. [2] Among long-distance migrants, this is typically achieved in sequences of directed daily or nightly flights (hereafter, flight-steps), interspersed by periods of extended stopover (hereafter, stopover). [3] Naïve migrants are thought to accomplish such feats by following inherited migratory directions, re-determined at the onset of each flight-step using various geophysical migratory compasses.”

R1.7 L11-12: the star compass model most favoured does not require time compensation. Authors discuss the role of the time-compensated model in the discussion but might be improved by being more explicit in both places (intro/discussion).

We agree with this excellent suggestion. In the Introduction, we now describe (P3 L22-24) the current lack of support for the time-compensated star compass, as well as how the time-compensation for the star compass and sun compass differ (given birds and insects are thought to respond to sun azimuth rather than the sun’s arc). We further report (P6 L19-21) as suggested by reviewer #2, that migratory self-correction based on time-compensation to the 15°/h star-rotation has previously been considered in the literature (but not how angular speed in sun azimuth requires a different response). Finally, we now mention in the Discussion (P32 L8-10) that, given the apparent flexibility of self-correction based on responses to sun-azimuth (e.g., Fig. 7), time-compensation using a star-compass could potentially be achievable in some way not diagnosable via clock-shift experiments (*sensu* Guilford and Taylor 2014).

R1.8 L20-24: an example of a sentence that is difficult to understand: “(1) geographic loxodromes, following constant headings relative to geographic South or North, which is identifiable either by a primary star compass^{7,9} or else by averaging (more reliably available) maximum bands of polarized light at sunrise and sunset^{10–12}”. The clarity might be enhanced if the geometry of the expected trajectory (constant heading relative to geographic north) is separated slightly from the mechanisms that might provide this. Throughout the introduction, the authors attempt to give all the information at the same time which makes it really tricky to follow!

We agree and have expanded the Introduction throughout. We now as you suggest separately describe the proposed compass cues in a single paragraph (P3 L16 - P4 L13), followed by a second paragraph which describes choice of primary compass and resultant compass courses (P4 L15 - P5 L17). The first paragraph includes how migrants

can potentially identify the geographic axis of the star compass or polarized light (P3 L22-24 & P4 L11-13), and the geographic loxodrome is now described in the second paragraph (P5 L1-3).

In the Introduction, we now also more carefully describe naïve migratory self-correction in a separate paragraph (P6 L16-24), rather than in a single sentence (originally P4 L10-12).

Page 4

R1.9 L1-5. I'm not sure that following a constant heading relative to sunrise or sunset azimuth is really menotactic, in that by paying attention to the sun's azimuth only at dusk and dawn it is quasi-time compensated. Perhaps useful here would be to consider time-limited compass information and whether that might be sufficient given an appropriate heading indicator (for semantic distinction see Guilford and Taylor 2014).

Thank you for the helpful suggestions. We have removed "menotactic". We also now mention the possibility of a time-limited compass (P4 L5-6) and possible advantage of a time-limited sun compass during twilight (P4 L6-9), during which the (hourly) speed of sun azimuth remains nearly constant through the year (Alerstam and Pettersson 1991).

Page 25

R1.10 L10-16: I think this is a very nice finding of the paper, but it would be lovely to have the authors write out in words in the main text their TSCS self-correction model in the introduction or here (and other key models, perhaps choose the ones that are discussed the most) rather than simply sending the reader to the equation and methods.

We agree – as you also point out, given the journal format – that the manuscript would benefit from more explicit descriptions within the main text regarding the key hypotheses and model components, rather than only in the Methods.

As detailed above, we now describe candidate mechanisms of self-correction in the Introduction, including via the star compass, and describe how this differs compared with the TCSC (P6 L16-24).

Additionally, we restructured the next paragraph to describe our model simulations more completely at the end of the Introduction (P7 L1-23).

We also include simple equations in the Results for more direct comparison, including how error-augmentation by spherical geometry effects directly arises from the equation for flight-step longitude (Eq. 2, and P8 L14-17). Also, for comparison with the cuckoo translocation experiment, we now include the simple equation governing TCSC corrections (Eq. 4, P11 L19-22).

We are pleased that you appreciate the close match between our model prediction and the observed shift in headings of the displaced cuckoos. We now also, in light of reviewer #2's comments, discuss alternative mechanisms of naïve migratory self-correction, including "emergency plans" when over water (P32 L17-19), and that the cuckoo displacements were also consistent with a time-compensated star-compass (P32 L22 – P33 L2).

Reviewer #2 (Remarks to the Author):

R2.1 This is a most interesting and important manuscript. Having been involved in quite some modelling/simulation work in the past, I very much welcome the authors' efforts to extend previous simplified migration modelling/simulation in the plane to the sphere, to using compasses associated with the rotating earth and to cue-transfers. It also means that I will necessarily be comparing with work of my own and I apologise in advance for referring more to my own work than I like or what it may deserve. Also, I have had some trouble in getting access to the relevant literature because of current corona restrictions.

We are very pleased that you appreciate our approach and contributions, and thankful for your extensive feedback and suggestions. We should in fact apologize for not adequately presenting findings from previous studies, including some of your own, and agree that doing so can provide a more complete picture regarding both the feasibility and limitations of “clock and compass migration” and compensations for displacement.

R2.2 The work allows for a revised perspective on the consequences of the use of different compasses. However, I am less convinced about the authors' inference based on their modelling/simulations. The uncertainties in the underlying model parameters are large complicating making firm conclusions about the compasses used. Perhaps more important, congruence between simulations and observed patterns can, at the conceptual level, mainly lead to conclusions, such as “a simple mechanism can lead to the observed patterns”, that can then be sought further validated/tested. Lastly, I do not believe that the authors present evidence that compensation observed in naïve migrants can be explained by enhanced migration performance as the authors seem to claim. I should add that the manuscript includes extensive material that cannot be thoroughly reviewed in the short time allowed by the journal. Thus, I have only critically examined what I believe are key (biological) issues and have assumed that the formulas in the method section are correct/meaningful.

We agree that our model and analyses provide more of a revised and extended framework rather than firm conclusions regarding compass course feasibility or cue favourability, as well as migratory self-corrections. We acknowledge that our use of terms like “explanation” was not properly qualified in this regard.

We have revised the manuscript to clarify this perspective. In the Introduction (P3 L12-14 & P5 L19-20) we state that the feasibility of compass-based long-distance migration remains an open question. In the Discussion, we now describe other factors affecting choice in compass cues (P30 L21 – P31 L5 & P33 L3-6) and possible alternative explanations for enhanced naïve migration performance (P33 L8-13).

To emphasize that model simulations are not so much diagnostic regarding feasibility of specific routes as regarding key global determinants of feasibility and cue hierarchy, we have revised and moved the summary Figure of global performance for a generic migrant from the (originally Suppl. Fig. S6) to the Results (now P18 L16-24, Fig. 5). This Figure illustrates how compass precision limits feasible longitudinal ranges among courses (Fig. 5a-c), and further impacts the relative performance gain with TCSC use over loxodromes (Fig. 5d-e).

Nonetheless, we would argue that our analyses and model results show that a sun compass is the only compass course compatible with both the longest-distance species routes and with self-correction (P31 L16-22). In short, as you write, there is good

evidence as you write that “a simple mechanism can lead to the observed patterns” (though the questions of sensory mechanisms and sufficient precision remain uncertain).

Also, we now both highlight and qualify our regression and model selection analysis (P27-28, Fig. 8), by stating that “While naïve migratory performance is primarily constrained by compass precision and goal breadth... we propose that it is further modulated [by] the number of required flight steps, [a] spherical-geometry factor and ... and flight-step distance.” (P30 L3-7). To illustrate the role of precision and latter “performance gain factors” in tandem among species (P28 L4-6), we also now provide a new Supplementary Figure (Suppl. Fig. 5).

Please see our replies to R2.6-2.7 where we discuss the issue of enhanced migration performance,

R2.3 An important theme is the potential correction/apparent compensation caused by the use of a time-compensated sun compass without resetting of the internal clock (originally called pseudo-navigation by Rabøl 1998) or with compass course transfer in combination with resetting of the clock. It is not really a new idea – for instance it was discussed extensively in Thorup and Rabøl (2007). It is of course of considerable interest to model/simulate the effects along migration routes.

We agree that we could have both more clearly described previous studies addressing naïve migratory self-correction, and also explained how our approach and results differ from earlier work. We of course agree that the proposal that juveniles self-correct is indeed not really new, as we now summarize in a separate paragraph in the Introduction (P6 L16.24).

However, despite extensive searching, we failed to find a description of self-correction using the sun compass in the literature. Emlen (1975) describes self-correction following clock-shifts extensively based on the 15° hourly rotation of the stars but does not point out the difference using sun azimuth. Alerstam and Pettersson formulate shifts in (error-free) headings for a non-reset sun azimuth compass, but do not mention self-correction for errors *en route*. And in your extensive treatment of experimental evidence for self-correction in Thorup and Rabøl (2007), self-correction using a sun compass is not mentioned, except in stating that the azimuths of the sun and stars move at “about 15° per hour”. We therefore now contrast tracking the 15° hourly rotation of the stars (P3 L22-24) with tracking sun azimuth across the horizon (P3 L24 – P4 L9), the latter involving variable hourly angular adjustments.

R2.4 But it is not made clear to what degree the “new” conclusions result from use of different parameters or inclusion of additional components. Indeed some previous work exists and I was surprised to not find it more directly cited and compared to the presented work (for example simulation of marsh warbler migration in Thorup et al. 2007).

See our reply to R2.2 where we

We agree that a more direct comparison of previous simulation studies would be helpful, and apologise in particular for having overlooked your 2007 Marsh Warbler simulations. To this end, we have added an uncertainty analysis of the Marsh Warbler (P23, and Fig. 6), illustrating the combined effects of compass precision, goal radius (from 100-1100 km) and between-individual variability (0°-10°, i.e., $\kappa = 33$, circular length = 0.985).

See our reply to R2.17 (comment to Table 2) where we discuss how this compares with your Marsh Warbler analyses in more detail, and possible consequences for compass course feasibility.

R2.5 Also, previous literature on a similar subject has used slightly different annotation. You might of course have good reasons for choosing a different one but providing some reference to previous work would make comparison more easy. Variation in migration direction is in most work incorporated as the concentration of a von Mises distribution but here as effective step-wise error. It would help the reader (and the reviewer) to know how these two compare. In the present work, effective standard errors of 2.5deg in between-individual headings are used but how does that compare to an r-among of 0.98 where 95% of the angles fall inside +/- 22deg of the mean angle? Did the authors include less variation among individuals than in Thorup et al 2007? (I do not have access to the literature to find out myself – but I also think the authors should provide this information).

We did in fact use von Mises headings throughout the study, which we now state more clearly (P10 L1-3). As in the original submission, to facilitate interpretation among those less familiar with circular statistics, we also report variability in orientation in degrees based on an “effective error” derived from von-Mises concentration (P10 L4-7). However, we acknowledge that the term “effective error” could lead to confusion, so therefore now describe orientation error in terms of “directional precision among daily or nightly flight-steps” (P7 L16) or, when assessing within-flight-step processes, as “compass precision” (P2 L7, P7 L17-20). For better comparison with previous studies and for those familiar with circular statistics, we now additionally list von Mises concentrations and mean vector (circular) lengths (when not being repetitive).

You are right, our “default” between-individual variability of 2.5° ($\kappa = 525$, circular length = 0.999) is less than is typically estimated from cage experiments, e.g., with only 3 of 16 experiments listed in Thorup et al 2007 with circular length > 0.99. However, in our uncertainty analysis (P23, Fig. 6), we now explore a broader range of between-individual variability (up to 10°, i.e., $\kappa = 33$, circular length = 0.985), more consistently with Thorup et al 2007.

R2.6 Re the suggestion/conclusion that the self-correcting sun compass can explain observed responses to displacements, they make a case only for the cuckoo displacements – obviously, the longitudinal displacement of 14deg in the Faroes displacement (Thorup et al 2011) does not render clock-shift a likely explanation of the much larger changes observed (68deg). “For the perhaps clearest and most convincing case for naïve self-correction, by GPS-tagged Eurasian Cuckoos (*Cuculus canorus*) following a 28deg longitude displacement at 55degN 20, the estimated shift in headings compared with 15 non-displaced “control” individuals (21deg) is intriguingly close to as predicted (23deg) using Equation (21)”. However, with a 28deg longitudinal change the predicted clock-shift change is 28deg – the 23deg is the change expected if birds fly themselves around sunset/sunrise without resetting; the 21deg though is still not significantly different from this. But more important, accepting the time-compensated sun compass with inner clocks being updated and last headings maintained during stopover, the routes should curve southward and the with the observed angle after 500 km the birds would not reach the normal migration route as observed in the study (where the change in direction was on average kept throughout).

You are of course right regarding discrepancy with the Faroe Island displacement data. We now mention (P32 L17-19) that these could represent an emergency plan for Western European migrants when lost at sea, e.g., as proposed by Mouritsen (1999,

2003). (We could add that the range of departure directions from the Faroe Islands was extremely broad, with many Easterly corrections coinciding with strong Westerly winds. Nonetheless, a fascinating and compelling study).

Regarding the cuckoo-experiment corrections, we now mention (P32 L22 – P 33 L2) that they were also compatible with a star compass and – given the seeming flexibility of TCSC to how clocks are updated and sun azimuth adjustments retained – that other kinds of responses to stars or sun azimuth could potentially produce partial self-corrections.

It is however not clear to us that the “corrected” flight directions among translocated *juvenile* cuckoos (Fig. 1b-c in Thorup et al 2020) were in fact maintained on average, e.g., the two cuckoos tracked farthest are in fact more suggestive of increasingly Southerly headings (though of course we cannot tell whether truncation of the others was due to mortality or instrument failure). Nonetheless, we agree that we cannot claim that a TCSC course explains the entire ‘normal’ migration route of juveniles, particularly since they eventually cross the Equator, which we have not addressed, and secondly because they seem to initially head directly South and not SW, which as you mention would be expected with an pre-fall-Equinox TCSC course. (However, we have modelled the Northern Hemisphere part of this route, and with moderate, e.g., 20-degree precision among steps, the SW ‘curve’ is not readily discernible.)

Overall, we do agree that other orientation tactics could result in direction changes following large displacements, such as threshold or signpost responses to the geomagnetic field, which we now discuss (P 33 L8-13).

R2.7 The main argument against ‘pseudo-navigation’ has been that the birds had ample time to reset their internal clock allowing birds to recalibrate whatever their preferred direction is and making it independent of the days before. In principle, a bird could arrive after displacement at a new location, note landmarks for their preferred (previous-location) time-compensated sun compass direction, and then transfer this new direction to their new preferred sun compass direction when they take off days after (which is different from a bird continuing in a direction it has flown on its own). I find this quite odd and without much evidence to support it.

We welcome the discussion about when birds reset their internal clock during migration. To the best of our knowledge, there is no hard evidence when they do this. However, as we mentioned (now P5 L14-15), sun compass experiments (summarized in Schmidt-Koenig 1990) revealed that full resetting clock shifts could take several days in pigeons and starlings. However, we agree that these hypotheses and early experiments would need very careful re-examination, particularly in the context of migration.

We also agree and now mention (P25 L14-15 & Methods P38 L13-15) that the proposal of Alerstam and Pettersson (1991), that migrants can maintain TCSC courses during stopovers by maintaining their (geographic) flight direction on landing, is neither self-evident nor in fact completely self-consistent with “transient” TCSC migrants, which effectively ignore their geographic heading once landed (by following a pure sun compass heading the next night). This effectively implies that a migrant somehow chooses on arrival whether it will be stopping over for an extended period or not, which is most unlikely considering the multi-faceted function of stopovers (Schmaljohann et al. 2022, Biol Rev).

We now, in addition to our simulations based on the “maintaining geographic-headings” TCSC model, suggest and formulate (P25 L15-18 & Methods P38 L15-20) a more parsimonious behaviour, which leads to nearly identical results (updated in Fig. 7c-d, Suppl. Fig 3). Namely, a migrant *always* flies relative to its TCSC heading, but only as experienced on the *first night of stopover*. This covers both transient and extended stopovers, and does not require migrants to predict their future nor ignore the sun while stopping over.

R2.8 I believe there is little evidence to suggest that migrants in general follow great circle routes apart from in very specific settings (such as in Alerstam et al 2001 without resetting presumed or Thorup (1998) Vagrancy of Yellow-browed Warbler *Phylloscopus inornatus* and Pallas's Warbler *Ph. proregulus* in north-west Europe: Misorientation on great circles? *Ringing & Migration* 19:7-12).

We do not completely agree, at least among long-distance routes. For example, Hobson and Kardynal (*The Auk*, 2015) found Western Veeries to follow near great circle routes, and Blackpoll Warblers tagged in Alaska and NW Canada likely also followed near great circle routes (unexpectedly passing over Georgia and Florida rather than as generally assumed North of Cape Hatteras; DeLuca et al, *The Scientific Naturalist*, 2019). Any detoured or multi-leg route (notably among Palearctic songbird migrants) will of course also deviate from great-circles.

Also, the often-cited notion that TCSC courses follow great circles is actually virtually untested. As far as we can tell, only Muheim et al (*Mvt Ecol*, 2018) actually compute the sun azimuth rather than use Alerstam's approximation of 1° shift in heading per degree of longitude displacement (Sokolovskis et al, *Mvt Ecol*, 2018, in Fig. 2C seem to additionally subdivide the latter to approximate partial clock resetting en route). Our simulations of the *yakutensis* Willow Warbler demonstrate that when accounting explicitly for sun azimuth and stopovers, the incremental changes due to shifts in sun azimuth with latitude and between nights through the season (Eq. 9, P36 L11) result in slightly less Northerly headings being taken than when assuming an exact great circle.

But we do certainly agree that not all long-distance trajectories necessarily represent single-heading TCSC courses! (For example, for Alaskan-born wheatears, a *Zugknick* heading or coastal response would most likely be required.)

R2.9 And, I have difficulties envisioning how it [TCSC courses] should work with the transfer of daytime cues for the translocated migrants where the clock is likely to be maintained more by light-dark cycles than actual sun azimuth changes.

This is also a good point; it is indeed unclear how (putative) TCSC migrants would update their clocks and responses to sun azimuth, as we now mention (P32 L8-9).

However, if dark-light cycles trigger inner clock resets, the classic clock-shift experiments reviewed in Schmidt-Koenig (1990) seem to indicate that clock resetting still takes several days. This could therefore also be the case for both naturally and experimentally displaced migrants, consistent with our model assumptions.

Specific comments (random):

R2.10 P215: "airborne compass courses" – what is that?

Changed to "... modelled robustness among compass courses for diverse airborne migrants" (P2 L5-6)

R2.11 P217: "previously overlooked spherical-geometry" – were they overlooked or have people just not considered navigation on a sphere? Perhaps just delete previously overlooked.

We now omit the term "previously-overlooked".

Nevertheless, as far as we can find, the augmentation of orientation errors in compass-based movement through the convergence of meridians is truly overlooked in the migration literature (as we mention in P30 L12-14). As far as we can ascertain, it is also rarely mentioned in the human navigation literature. This could be related to human navigators going beyond consideration of simple compass courses by the 16th Century (as we mention in P30 L16-18), through developing (albeit imprecise) maps and soon afterwards Mercator projections.

R2.12 P2111: "increasing goal-area" – can an area increase?

Now changed to "increases with larger goal areas..." (P2 L11)

R2.13 P2112-14: "Our results can explain enhanced naïve migrant performance, observed diversity in compass-cue hierarchies, and sun-compass orientation being key to many long-distance inaugural migrations" – there is something odd with the sentence structure combining "can explain" with "being key". Furthermore, as outlined above I find the conclusion too far-reaching.

Agreed. Added qualifications "can help explain [enhanced naïve performance]" and "support...being key" (P2 L12-13).

R2.14 P315 onwards: there is little distinction between what birds have been shown to be capable of (typically in cages) and what they actually do when migrating (ie. free-flying).

Thank you for pointing out that this was not sufficiently qualified. We now open the paragraph describing compasses (P3 L16 onwards) with "Based mainly on captive individuals, naïve but migration-ready birds and insects". Additionally, in the next paragraph (P4 L18-24), we emphasize (P4 L20-24) that cue-conflict experiments have been restricted to departure decisions.

R2.15 P416: you might want to mention that this conclusion was questioned – see the debate Mouritsen (1998), Thorup et al (2000), Mouritsen (2000) in Animal Behaviour.

We did not mean to ignore or side-step this question, and now dedicate a paragraph to describe the knowledge gaps regarding inaugural routes (P5 L19 – P 6 L 14), including that "The extent to which compass courses can result in successful migration routes remains an open question. Central to this question is whether cue perception, compass headings and resultant flight-step directions are sufficiently precise" (P5 L19-20).

R2.16 P5 results: I found it very difficult to understand the results without extensive reference to the methods section to find out what has been done – perhaps provide some more information either in the results section or the last part of introduction to assist interpretation.

This is in agreement with the first reviewer (R1.2-1.3). As outlined in our responses to them, we have revised both our Abstract and Introduction to more clearly state our context and aims. In particular, we now outline which factors are assessed (P7).

In the beginning of the Results, we now include a heuristic explanation of how spherical-geometry effects enhance errors at high latitudes (P8 L14-17), how headings were measured (P9 L1-9) and more carefully describe how performance in the normal limit varies with the length-adjusted goal breadth (P10 L17-20).

R2.17 Table 2: why use a different goal radius than Thorup and Rabøl 2007 which you cite? The 330 km step length also deviates substantially from previous estimates. Mouritsen (1998) estimated r-step as 0.665 for ring recoveries 100-150 km distant – how does that compare to the 20deg effective step error and 300-360 km step-wise distance you consider?

We indeed used a larger goal radius for the Marsh Warbler simulations (500-km) compared with 100-km in Thorup and Rabøl (2001) and Thorup et al (2007) which, as we now explain in the Methods (P42 L2-4), effectively assumes that migrants can use other cues within a shorter “post-long-distance scale”, e.g., *sensu* Mouritsen (2018) and Bingman and Cheng (2005).

Regarding directional precision among flight-steps (“r-step”), we now clarify (P9, Table 2, P17 L3-4) that we tested directional precision among flight-steps within 60° (r-step ~0.4).

As mentioned above (R2.4), we now include a more in-depth parameter uncertainty analysis for Marsh Warbler migration (P23 and Fig. 6) regarding not only compass precision (0°-40°), but also goal radius (100-1100 km) and between-individual variability up to 10° (concentration 0.985).

For parameter estimates consistent with Thorup et al 2007, i.e., between-individual variability with circular lengths 0.98-0.99 (8°-11.5°), and a 100-km radius, with 20° among-step variability (roughly twice as precise as Mouritsen’s estimate), Fig. 6d indicates 5% arrival following a loxodrome, and from Fig. 6e, about 7.5% with a TCSC.

We speculate however that, in addition to birds being able to pinpoint appropriate habitat once closer to goal areas, that the concentration among truly compass-based migratory flight-steps could be quite a bit higher than the early EURING analyses suggest. For example, from Supplementary Fig. 2, with 500-km and 2.5° between-individual variability, performance becomes 61% for a geographic loxodrome and 59% for a TCSC, respectively.

In the Discussion (P30 L21 - P31 L2), we now qualify our results given the remaining uncertainty of compass precision and directional precision among flight-steps. We believe that trying to resolve this specific case is not central to the scope and purpose of this study, which identifies and assess key factors governing compass course robustness, and is of course more predictive rather than diagnostic.

R2.18 P23I2: “first global assessment of factors governing robustness” - Apart from the unnecessary “first” you don’t seem to assess the factors governing robustness but rather assess whether the factors involved govern robustness.

We now drop the terms “first” and “factors governing” (P30 L1-3).

R2.19 P2516-7: how can refs 9+29 support that something proposed in ref 45 is "now generally regarded as not supported" when ref 45 is published after 9+29?

Revised to both emphasize (P32 L1-2) that while there is no direct evidence of the time-compensated star compass, it "is plausibly achievable in some way not diagnosable using clock-shifts". We also mention (P32 L22-23) that the cuckoo displacement results are in fact also compatible with a time-compensated star compass.

R2.20 P30 eq12-13: N_0 defined in (12) does not seem to be the one inserted (13)

Thank you for pointing out this typo, now corrected (and also now moved to Supplementary Information, P3 L15-20). The second half of the equation used an earlier definition of N_0 , not used in any of our analyses. We took N_0 to be the minimum number of steps to arrive at the closest edge of the goal area, as described (Supplementary Information, P3 L17). Apologies for the confusion.

R2.21 P37123: what is the concentration of a von Mises distribution with 2.5deg effective standard error?

We now list this value, 0.999 (P17 L11), but test other values as described in R2.17

Kasper Thorup

Reviewers' comments:

Reviewer #1 (Remarks to the Author):

Apologies for the long time to write what in the end is a very short review, the request came at the start of extensive fieldwork.

The MS is much improved and the clarity is now much better throughout. The exception is the abstract which I still feel doesn't quite sell the paper, or highlight the same key points as the discussion and so I wonder whether the authors might consider trying to make sure that the abstract, intro and discussion are coherent with each other. E.g. The phrases like "we found (i) sun-compass courses partially self-correct between flight-steps" makes it hard to understand the modelling approach.

Some v minor comments:

Page 4, Line 7: missing 'the' before sun azimuth

Line 11: should read 'which is'

Page 30, line 5: "further and modulated"?

Reviewer #2 (Remarks to the Author):

This is a resubmission of a manuscript that I have previously reviewed. I don't have time now for a full re-review so I will be evaluating mainly their response to my comments.

The authors have responded constructively to most reviewer comments and the overall restructuring has made it easier to read and understand. The authors present welcome simulations of routes extending to a sphere and include great-circle routes. I wish though that, when discussing the implications, instead of discussing a narrow selection of the literature the authors would present what would be critical (but lacking) future tests. Overall, the response leaves me with the feeling that the evidence for and implications of their simulations are overstated. The behavioural and experimental evidence for the necessary underlying behaviours – such as a general reliance of migrants on a time-compensated sun compass course adjusted at each new stopover – is scant/lacking. Reservations are few and far between and will most certainly be overlooked by most readers though identifying limits/limitations are equally important for scientific progress.

I very much welcome a discussion of explanations for the responses to displacement by young cuckoos but before citing it as support for self-correcting TCSC courses, some detail that appear to have been overlooked should be considered and some additional analyses might be worthwhile. The 21° change in bearings of displaced birds, cited by the authors as "intriguing close to as predicted", is and was not evidence for compensation/correction. In fact, the bearings after 500 km were not significantly different from those of control birds (as also reported in the study) and the bearings after 100 km were less, "only" 196°, despite the expectation of the strongest response early. The evidence presented is based on a continuation of the westward movement during the autumn migration.

Given that the short-term change is not different from the controls (and was even less after shorter distances) an evaluation of whether "TCSC routes with cue transfer at each stopover" could explain the observed patterns compares observed with expected routes over longer distances (although such a comparison can still only rule out an alternative, not prove it). The authors maintain their position based on their observation that it is not clear "that the "corrected" flight directions among translocated juvenile cuckoos (Fig. 1b-c in Thorup et al 2020) were in fact maintained on average, e.g., the two

cuckoos tracked farthest are in fact more suggestive of increasingly Southerly headings". Obviously, the small sample obviously makes evaluation difficult and I don't have the means or time for simulating expected outcomes. Nevertheless, a remarkable property of the TCSC dependent routes is that they curve (counter-clockwise for southwest courses). Although, the authors in their response claim a counter-clockwise shift in two cuckoos, a quick test of the cuckoo (based on direction from release site to locations) shows that routes on average turn clock-wise, and with at the individual level 5 of 7 cuckoos showing this. Such turning, and I hope the authors will agree, argues against the cuckoos following a TCSC dependent route. In any case, the individual variation is large and I think the most fair interpretation is that most of the observed variation could be explained by "simple" responses but that some individuals exhibited responses that are not easily explained by a "self-correcting" mechanism.

The authors have added the Faroe displacement as an example of a response that cannot be explained by "self-correction" but at the same time invoke an ad hoc explanation as the most likely one for the responses of the displaced birds which appears more to me as an attempt to explain experimental responses within a "self-correcting" framework than an unbiased evaluation. At least the experimentally displaced birds did not "find themselves over water" (they were well within cages) which should lead to executing the emergency plan. Furthermore, such ad hoc explanations might of course be true but they are next to impossible to evaluate scientifically. For instance, the emergency plan that the authors mention would render many migrants poor of if they invoked the same "emergency plan" in the Mediterranean. And, the evaluation of many displacement experiments by Thorup and Rabøl pointed to a time-compensated star compass not being a sufficient general explanation though the star-compass version leads to larger expected corrections the sun-compass with cue transfer.

Furthermore, it is relevant to note that given the proposed widespread reliance on TCSC with cue transfer at stopovers, a lack of correctional response is also unexpected. The juvenile white-crowned sparrows in Thorup et al. 2007 did not correct, yet, they migrate on a curved route that curve similar to expectations from TCSC. Neither did Perdeck's juvenile starlings appear to correct.

Re the paragraph on "The finding that TCSC courses are self-correcting provides a novel explanation for naïve migrants mitigating orientation errors and making route-corrections following displacements", I maintain that "self-correcting" mechanisms are not a novel explanation as also acknowledged by the authors otherwise though only (very briefly) summarised in the introduction.

For the authors' information, the common name of the species that was displaced is common cuckoo – not Eurasian Cuckoo – and, though I would have loved it to be higher only eight juveniles were displaced. And, of those eight juveniles only 2 were GPS-tagged, the others were tracked by satellites. A minor thing: "Self-correcting sun compass" as stated in the title is, I believe, slightly misleading. Isn't it the course/route that is "self-corrected"? And is "self-correcting" not used differently in p33115, at least when the Faroe displacements are included.

Reviewers' comments:

Reviewer #1 (Remarks to the Author):

Apologies for the long time to write what in the end is a very short review, the request came at the start of extensive fieldwork.

R1.1 The MS is much improved and the clarity is now much better throughout. The exception is the abstract which I still feel doesn't quite sell the paper, or highlight the same key points as the discussion and so I wonder whether the authors might consider trying to make sure that the abstract, intro and discussion are coherent with each other. E.g. The phrases like "we found (i) sun-compass courses partially self-correct between flight-steps" makes it hard to understand the modelling approach.

We are naturally very pleased to hear that our revisions have improved the MS and its accessibility. We agree that the Abstract was however not updated effectively. We have now revised the Abstract to include the specific approach used for each of the main findings, e.g. (P2 L4-10),

"To assess compass-based naïve migratory performance (successful arrival), we modelled and simulated proposed compass courses for diverse airborne migratory populations...We formulate how time-compensated sun-compass headings partially self-correct, according to how inner-clocks are updated. For the longest-distance migrations simulated, time-compensated sun-compass courses were most robust to error, and most closely resembled known routes."

Some v minor comments:

R1.2 Page 4, Line 7: missing 'the' before sun azimuth

Done (P4 L9)

R1.3 Line 11: should read 'which is'

Done (P4 L13)

R1.3 Page 30, line 5: "further and modulated"?

This was a typo (should have read "further modulated" – now reworded (P30 L9-10).

Reviewer #2 (Remarks to the Author):

R2.1 This is a resubmission of a manuscript that I have previously reviewed. I don't have time now for a full re-review so I will be evaluating mainly their response to my comments.

We appreciate the reviewer's detailed response and concerns. We have further revised the manuscript accordingly, as detailed below.

R2.2 The authors have responded constructively to most reviewer comments and the overall restructuring has made it easier to read and understand. The authors present welcome simulations of routes extending to a sphere and include great-circle routes

We are glad to hear that our responses were generally constructive and clarifying.

R2.3 I wish though that, when discussing the implications, instead of discussing a narrow selection of the literature the authors would present what would be critical (but lacking) future tests.

If we understood correctly, the reviewer here first mentions a perceived over-focus on compass-course performance (including literature discussed), and secondly that they would prefer that we additionally suggest possible experimental tests to distinguish time-compensated celestial orientation from alternative possibilities.

To address the first point about implications – which the reviewer reiterates in R2.4 – we have revised the manuscript to more clearly qualify our results as being restricted to compass-based naïve migratory performance, and better highlight potential extensions to and possible alternatives to single compass courses:

- (i) In the Abstract, instead of describing how “our results can explain... performance”, emphasize (P2 L12-13) that our study is predictive and focused on compass-based migration:
“Our predictive study provides a basis for assessment of compass-based naïve migration and mechanisms of self-correction...”
- (ii) In the first paragraph of the Discussion (P30 L1-4), we drop the words “explanation” and “enhanced” when describing that “[our formulations have] provided an explanation for enhanced performance”. We now state that
“Our extended formulations have facilitated a global assessment of robustness among compass courses, providing a predictive framework of naïve migratory performance and compass cue favourability among airborne migratory species and routes.”
- (iii) Regarding our revised presentation and qualification of the TCSC and self-correction in particular, please see our replies to the reviewer's comments R2.4-2.9, all of which relate to the TCSC.
- (iv) We have revised the Discussion to emphasise the possibility of “naïve navigation” abilities, e.g.,
“Moreover, the extent to which naïve migratory performance involves non-compass (e.g., navigational) mechanisms remains to be resolved.” (P30 L15-16)

“An important consideration is whether compass-based movement can accommodate such spatiotemporally variable factors, as well as in the Earth’s geomagnetic field, without requiring more sophisticated (naïve navigational) abilities.” (P33 L18-22),

- (v) We now more fully describe (P34 L3-11) such proposed abilities, regarding use of geomagnetic gradients by naïve migrants:
“Alternatively, more advanced naïve migrant abilities beyond compass-based movement have been proposed to explain enhanced orientation correction following displacement, or control of naïve trans-oceanic migration routes. Naïve migrants are accordingly proposed to gauge gradients in both geomagnetic intensity and inclination along their inaugural route, to either adjust (inherited) compass headings as a corrective measure, or else perform gradient-based navigation towards (inherited) geomagnetic goal signatures. Apart from perceptive and cognitive feasibility, the efficacy of the former ability and the efficiency of the latter remain to be established, in particular given the overall N-S gradients in both geomagnetic intensity and inclination.”,
- (vi) Lastly, regarding literature cited, in addition to the 9 already-cited references which consider or promote naïve navigational abilities (11-14, 42, 49-51, 77), we now additionally cite three reviews of migratory bird navigation which consider this in more detail (45: Wallraff, Vogelwarte 1977, 75: Wallraff, in Berthold 2001, and 4: Holland, J Zoology, 2014), an additional study (70: Thorup et. al, PNAS, 2007) not showing compensatory movements among displaced juvenile songbirds, and two computational studies (78-79) which explore naïve (oceanic) navigation in non-orthogonal fields mimicking geomagnetic components.

Regarding the second point, we find it an excellent suggestion to propose critical tests and information to report in future studies. We now do so in the Discussion (P32 L20 – P 33 L2):

- (vii) “To diagnose possible involvement of celestial compass use through displacement of naïve migrants, we recommend carefully controlling for access to celestial cues throughout the study (to assess possible resetting of inner clocks). To help distinguish among inner clock, celestial and other cue effects, we further suggest displacing individuals from the same capture location to the East and West and, if possible, also clock-shifting locally-captured (i.e., non-displaced) migrants.”

R2.4 Overall, the response leaves me with the feeling that the evidence for and implications of their simulations are overstated. The behavioural and experimental evidence for the necessary underlying behaviours – such as a general reliance of migrants on a time-compensated sun compass course adjusted at each new stopover – is scant/lacking.

Reservations are few and far between and will most certainly be overlooked by most readers though identifying limits/limitations are equally important for scientific progress.

We appreciate the reviewer's critical thoughts, and as mentioned and outlined in R2.3, have revised accordingly to emphasize that compass courses may not offer a general explanation for naïve migratory performance.

Regarding the TCSC and its "necessary underlying behaviours", we would first like to point out

- a) Its self-correction requires no additional abilities beyond those assumed in previous studies. That is, the TCSC, as formulated by Alerstam and Pettersson (1991) and simulated in Åkesson and Bianco (2015, 2017) and Muheim et al (2018), would also all self-correct, if they had considered orientation errors (note in Alerstam et al 2001, TCSC courses were simply approximated by 1° shifts in heading per degree of longitude; as stated in their Fig. 1).
- b) Rather, with our extended TCSC formulations (Eqs. 12a-c), we explore and assess the uncertainty of TCSC courses through testing plausible assumptions regarding retention of rates of time-compensation *en route*, and inner clock resetting behaviour.
- c) To this end, our finding (P32 L2-4) that

"TCSC courses can be robust to variable scheduling of flight-steps and inner clock resetting, as well as how headings are retained during stopover (Fig. 7)"

is encouraging, particularly for our novel formulation (Eq. 12c, Fig. 7d, Suppl. Fig. 3), where flight headings are always set relative to sun azimuth, and then retained during further stopover, i.e., transferred to and again from a more constant orientation reference (note, as mentioned on P7 L15-17, that a similar process would be required prior to the onset of migration to ensure reasonable trajectories with a primary sun compass of *any* kind).

We nonetheless agree with the reviewer that direct evidence of a TCSC as primary migratory compass is lacking, including regarding retention of headings during stopover. We additionally agree that it cannot fully explain the evident compensatory movements in the Faroe displacement study or Thorup and Rabøl's (2007) meta-analysis study (as we discuss further in R2.6-2.9).

We therefore now further emphasize that our TCSC formulations

- (i) are neither confirmed nor definitive (P33 L5-7)

"Regarding cue mechanisms, it is not yet clear... which if any migrants respond to sun azimuth and reset their inner clocks consistently with a TCSC."
- (ii) as mentioned in R2.3(vii), should be tested via displacement experiments which control more completely for access to celestial cues and possible resetting of clocks (P32 L20 – P 33 L2)

- (iii) serve as a “potential” rather than “novel” explanation of self-correction (P32 7-9)
 - “The finding that TCSC courses are self-correcting provides a potential explanation for how naïve migrants mitigate orientation errors, but the mechanisms underlying self-correction by naïve migrants following displacement remain unresolved.”
- (iv) even if adopted, cannot sufficiently explain all available evidence of naïve self-correction (P32 L16-19)
 - “However, shifts in orientation among juvenile songbirds after being displaced 16° to the West from Denmark to the Faroe Islands exceeded those predicted by a self-correcting star-compass (or TCSC), as did estimated corrections from a meta-study of orientation in funnels following real and virtual displacement.”

R2.5 I very much welcome a discussion of explanations for the responses to displacement by young cuckoos but before citing it as support for self-correcting TCSC courses, some detail that appear to have been overlooked should be considered and some additional analyses might be worthwhile. The 21° change in bearings of displaced birds, cited by the authors as “intriguing close to as predicted”, is and was not evidence for compensation/correction. In fact, the bearings after 500 km were not significantly different from those of control birds (as also reported in the study) and the bearings after 100 km were less, “only” 196°, despite the expectation of the strongest response early. The evidence presented is based on a continuation of the westward movement during the autumn migration. Given that the short-term change is not different from the controls (and was even less after shorter distances) an evaluation of whether “TCSC routes with cue transfer at each stopover” could explain the observed patterns compares observed with expected routes over longer distances (although such a comparison can still only rule out an alternative, not prove it). The authors maintain their position based on their observation that it is not clear “that the “corrected” flight directions among translocated juvenile cuckoos (Fig. 1b-c in Thorup et al 2020) were in fact maintained on average, e.g., the two cuckoos tracked farthest are in fact more suggestive of increasingly Southerly headings”. Obviously, the small sample obviously makes evaluation difficult and I don’t have the means or time for simulating expected outcomes. Nevertheless, a remarkable property of the TCSC dependent routes is that they curve (counter-clockwise for southwest courses). Although, the authors in their response claim a counter-clockwise shift in two cuckoos, a quick test of the cuckoo (based on direction from release site to locations) shows that routes on average turn clock-wise, and with at the individual level 5 of 7 cuckoos showing this. Such turning, and I hope the authors will agree, argues against the cuckoos following a TCSC dependent route. In any case, the individual variation is large and I think the most fair interpretation is that most of the observed variation could be explained by “simple” responses but that some individuals exhibited responses that are not easily explained by a “self-correcting” mechanism.

We appreciate the extra information and the reviewer sharing their concern that the displaced cuckoo tracks do not resemble TCSC courses. We have now analysed the cuckoo data (<https://www.datarepository.movebank.org/handle/10255/move.1073>) and agree, unlike as we thought, that the data tend to show a clockwise directional shift.

We would argue however that such a shift is somewhat expected given the strong global wind patterns when heading South to Africa (whether displaced or not). This goes beyond the scope of our MS, but can be understood as follows

- a) The Figure below compares (left) the trajectories of displaced juvenile cuckoos (dark blue lines, taken from Thorup et al., 2020) with (right) “rose plots” depicting the distribution of winds aloft (850mb, i.e., ca. 1500m) during the period of post-displacement movements (September winds over Eastern Europe and the Middle East, and October winds over East Africa, taken from the Appendices in Kemp et al 2010, J Avian Bio). The added blue arrows on the right panel depict simplified cuckoo trajectories, for comparison.
- b) We found further evidence of prevalent drift (but do not share details here), by comparing the direction of incident wind conditions during the early movements of the displaced juvenile cuckoos with the resultant flight directions. For example, comparing with NCEP Reanalysis 1 winds (<https://psl.noaa.gov/mddb2/makePlot.html?variableID=156035>), we found that the individual with tag 157309, which travelled farthest to the South-East (see maps), moved during steady winds to the East (Aug. 26-27, and Sept 6-9, 2016), while the individual with tag 61851 travelled most strongly Westward across the Black Sea in strong Westward winds (which prevailed throughout Oct 9-22, 2018).

- c) Therefore, while we agree that the sample size (and our brief anecdotal analysis above) preclude firm conclusions, we argue that the cuckoo data neither necessarily rules out the use of a time-compensated sun compass, nor necessitates an alternative, e.g., navigational, ability.

We have therefore further carefully revised the manuscript in accordance with the reviewer's concerns by

- (i) emphasising in the Discussion, as detailed in our replies to R2.3(iv-v) and R2.4(iii-iv), that the mechanisms underlying naïve route corrections remain unclear,
- (ii) in addition to polar, equatorial and inner clock effects, add wind as a complicating variable when interpreting experimental evidence for route corrections (P32 L9-1)
 - “Interpretation of experimental evidence of naïve self-correction is often complicated by wind...”,
- (iii) summarizing the uncertainty of underlying mechanisms in this, the Faroe and the Thorup and Rabøl (P32 L19-20):
 - “The mechanisms underlying all of these corrections therefore remain unclear.”,
- (iv) adding suggestions (P32 L20 – P 33 L2), as described in R2.3(vii), to help unravel cue use and possible self-correction through displacement experiments.

R2.6 The authors have added the Faroe displacement as an example of a response that cannot be explained by “self-correction” but at the same time invoke an ad hoc explanation as the most likely one for the responses of the displaced birds which appears more to me as an attempt to explain experimental responses within a “self-correcting” framework than an unbiased evaluation. At least the experimentally displaced birds did not “find themselves over water” (they were well within cages) which should lead to executing the emergency plan. Furthermore, such ad hoc explanations might of course be true but they are next to impossible to evaluate scientifically. For instance, the emergency plan that the authors mention would render many migrants poor of if they invoked the same “emergency plan” in the Mediterranean.

We agree with the reviewer that this part was not carefully written. We have therefore now deleted the sentence mentioning “emergency” i.e., contingency plans. While we believe such plans may potentially play significant roles in avian migration, we agree that mentioning it is not particularly helpful here, and the “over water” contingency is inappropriate for funnel orientation.

Additionally, as mentioned in R2.4(iii), we now additionally cite the Faroe study as an example of a corrective shift in orientation exceeding that predicted by a TCSC.

R2.7 And, the evaluation of many displacement experiments by Thorup and Rabøl pointed to a time-compensated star compass not being a sufficient general explanation though the star-compass version leads to larger expected corrections the sun-compass with cue transfer.

Thank you for pointing this out. We agree that the data reviewed in Throup and Rabøl (2007) are indicative of self-correction beyond that predicted by pseudo-navigation (star or sun compass self-correction), especially among short displacements.

We therefore

- (i) more explicitly mention star-compass self-correction while citing Thorup and Rabøl in the Introduction (P6 L 19-22)
 - “The ability to correct orientation following longitudinal displacements is a hallmark of true navigation, but could also be achievable if

- migrants tracked configurations of stars or the sun in a time-compensated way (also termed *pseudo-navigation*).”
- (ii) as mentioned in R2.4(iii), acknowledge in the Discussion (P32 L16-19) that the meta-analysis is underpredicted by TCSC or star-compass “pseudo-navigation”

R2.8 Furthermore, it is relevant to note that given the proposed widespread reliance on TCSC with cue transfer at stopovers, a lack of correctional response is also unexpected. The juvenile white-crowned sparrows in Thorup et al. 2007 did not correct, yet, they migrate on a curved route that curve similar to expectations from TCSC. Neither did Perdeck’s juvenile starlings appear to correct.

We must admit that we might have misunderstood this comment. We never intended to suggest that TCSC should be ubiquitously adopted, since cue favourability will depend on migration distance and direction (P30 L8-13), relative compass precision and cue availability (P33 L3-16), as well as the possibility of multiple headings to accommodate environmental drivers (P33 L17 – P34 L3).

Also, given the cuckoo data, Throup and Rabøl (2007) and the Faroes data all seem to indicate an overall multi-species pattern of correction following displacement (see also our responses to R2.4-2.6), we dare to argue here that the apparent lack of correction by a single night-migratory species (*gambelii* WCSPs, not a solely long-distant migrant), and by short-distance socially-migrating starlings (which in Perdeck 1958 may have followed experienced conspecifics to Spain), is neither completely unexpected nor grounds to rule out possible TCSC self-compensation existing elsewhere.

We do agree that the WCSP study (Thorup et. al, PNAS, 2007) is relevant, and now (P32 L11-13) cite this as the only of three studies which tracked naïve migrants following displacement which did not reveal corrective orientation.

R2.9 Re the paragraph on “The finding that TCSC courses are self-correcting provides a novel explanation for naïve migrants mitigating orientation errors and making route-corrections following displacements”, I maintain that “self-correcting” mechanisms are not a novel explanation as also acknowledged by the authors otherwise though only (very briefly) summarised in the introduction.

We agree with the reviewer. As mentioned in R2.4(ii), we now replace the word “novel” with “potential” and qualify evidence of its potential regarding route corrections (P32 L7-9, L19-20).

R2.10 For the authors’ information, the common name of the species that was displaced is common cuckoo – not Eurasian Cuckoo – and, though I would have loved it to be higher only eight juveniles were displaced. And, of those eight juveniles only 2 were GPS-tagged, the others were tracked by satellites.

We are grateful for the extra information and corrections, and apologise for our carelessness in reporting. We have now corrected this information (P32 L13-14).

R2.11 A minor thing: “Self-correcting sun compass” as stated in the title is, I believe, slightly misleading. Isn’t it the course/route that is “self-corrected”? And is “self-correcting” not used differently in p33l15, at least when the Faroe displacements are included.

Technically speaking, we agree that it is the course which is self-correcting, but it is the compass that determines the course. So, we would say that it is not self-evidently either one or the other.

Rather than the term “self-correcting” being different between a TCSC and the Faroes, we would say it is the actual degree of self-correction and underlying processes that apparently differs. For clarity, we now reserve the term “self-correct” for TCSC (or pseudo-navigation), and use the terms “corrective” (e.g., P6 L20, P32 L8, P34 L7) or “corrections” (e.g., P32 L18, P32 L19, P34 L4) when describing estimated orientation shifts following displacement.

To further consider the reviewer's comment, and to help further prevent the misunderstanding that we claim the TCSC to be ubiquitous or “the norm”, we have adjusted the title to “**Naïve migratory performance over the globe: many-wrongs, cue transfers and sun compass self-correction**”. We hope that, worded in this order, the title better first emphasizes the study's focus and contributions, i.e., assessing naïve migratory performance on a spherical Earth, accounting for errors and convergence in orientation, and a presentation of the self-correcting properties and performance of the TCSC (while avoiding the ambiguous term “predict”).